# CLIP AS A PRIOR TEACHER: BREAKING THE LABEL DEPENDENCY IN SEMI-SUPERVISED LEARNING

## ABSTRACT

Semi-supervised learning (SSL) has shown remarkable potential in scenarios with limited labeled data. However, our study reveals that existing SSL approaches remain inherently label-dependent—their ability to exploit unlabeled samples is bounded by the quantity and quality of labeled data. To address this limitation, we establish a portable asymmetric-modalities co-training framework for efficiently integrating CLIP into SSL, termed CaPT. CaPT aggregates predictions from a fully fine-tuned unimodal network and a parameter-efficiently fine-tuned multi-modal CLIP model via carefully designed co-pseudo labels, which guide training by refining CLIP's biased predictions and supplementing reliable prior for SSL without compromising efficiency. Moreover, the asymmetric-modalities mitigates the pattern-homogeneity bottleneck observed in previous co-training methods, enabling richer cross-model information exchange. CaPT consistently achieves state-of-the-art performance across multiple SSL benchmarks. Notably, it outperforms the second-best method by **21.38%** and **4.05%** on the CIFAR-100 and EuroSAT datasets, respectively, under the one-label-per-class setting, demonstrating its strong potential in low-label regimes.

## 1 INTRODUCTION

Semi-supervised learning (SSL) aims to reduce the reliance of supervised training on large-scale labeled data. Recent advances in thresholding strategies (Zhang et al., 2021a; Wang et al., 2023) and generalization techniques (Berthelot et al., 2019b;a; Huang et al., 2023) have significantly enhanced SSL performance. Even in low-label regimes, SSL achieves promising results (Han et al., 2025). Nevertheless, we note that SSL methods still exhibit a heavy dependency on labeled data, with performance dropping sharply once label quantity falls below a critical threshold. As shown in the left subfigure of Figure 1a, SSL algorithms perform competitively on the CIFAR-10 dataset (Krizhevsky, 2009) with as few as 400, 25, or even 4 labeled samples per class. However, their performance deteriorates markedly when the labeled data is reduced to just one sample per class. In addition, the quality of labeled samples matters. Following prior work (Sohn et al., 2020), under the one-label-per-class setting on CIFAR-10, we construct three labeled training sets using the prototypicality ordering mechanism (Carlini et al., 2019): Set 0 contains the most prototypical image for each class, while Set 2 contains the least prototypical images. The radar chart in Figure 1a shows that SSL algorithms achieve the highest performance when trained on Set 0 and the lowest performance when trained on Set 2. Diving deeper, Figure 1b shows that, during FreeMatch (Wang et al., 2023) training, pseudo label accuracy is substantially lower when the labeled samples are less prototypical.

To complement these empirical observations, we present a analytic model and a supporting theorem. Under a prototype-based Gaussian-mixture generative model, let $g$ denote the minimum inter-class centroid distance, $\sigma^2$ the per-class noise variance, $B$ a uniform bound on the systematic bias of the chosen labeled prototypes (i.e., non-prototypicality), and $n_{\min}$ the minimum number of labeled samples per class. Define $\varepsilon_n := \frac{2\sigma}{\sqrt{n_{\min}}} \sqrt{\log\left(\frac{K\,2^{d/2}}{\eta}\right)}$, where $K$ is the number of classes and $d$ the input dimension. Let $r := B + \varepsilon_n$. we derive the following bound on the pseudo label error:

**Theorem 1.1.** *With probability at least $1 - \eta$ over the labeled-sample draws, the nearest-prototype pseudo label error for any class $c$ satisfies*

$$\Pr_{x|y=c} \left( \hat{y}(x) \neq c \right) \leq (K-1)\, 2^{d/2} \, \exp\left( - \frac{(g/2 - r)^2}{4\sigma^2} \right). \tag{1}$$

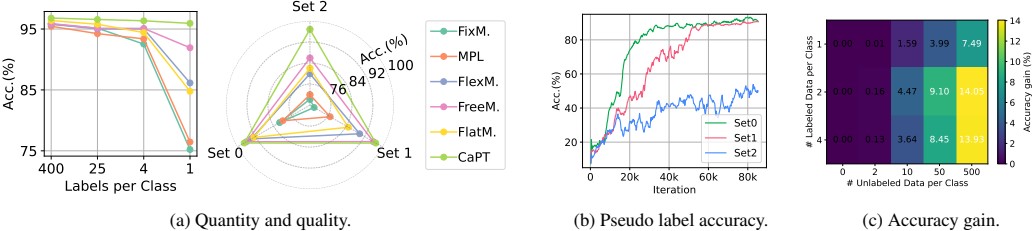

(a) Quantity and quality.     (b) Pseudo label accuracy.     (c) Accuracy gain.

Figure 1: Motivating example of CaPT. (a) Existing SSL methods exhibit significant performance degradation under restricted labeled data. (b) The quality of labeled data affects the accuracy of pseudo labels for unlabeled data. (c) SSL struggles to benefit from unlabeled data when labeled data are extremely scarce.

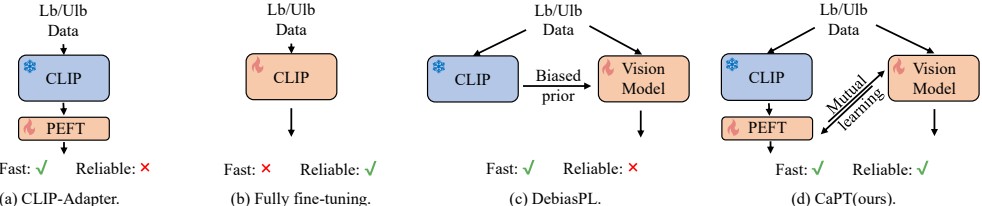

(a) CLIP-Adapter.     (b) Fully fine-tuning.     (c) DebiasPL.     (d) CaPT(ours).

Figure 2: Different framework for integrating CLIP into SSL. CLIP-Adapter and DebiasPL suffer from limited learning capacity and biased prior, resulting in unreliable predictions. In contrast, CaPT balances efficiency and reliability.

Consequently, the upper bound of the pseudo label error decays exponentially in $(g/2-r)^2/\sigma^2$ when $g/2 > r$. Increasing the prototype bias $B$, or reducing the labeled sample size (which increases $\varepsilon_n$), directly reduces the effective margin $g/2 - r$. A smaller margin in turn enlarges the upper bound of the pseudo label error, thereby substantially increasing the risk that the actual pseudo label error will rise. This reveals a fundamental limitation of existing SSL methods: *the utilization of unlabeled samples depends heavily on the properties of labeled data.* More concretely, although SSL ostensibly depends on two data sources, the utility of unlabeled samples is tightly **coupled** to labeled data. Paradoxically and unexpectedly, as the supervision from labeled data deteriorates, SSL instead becomes more dependent on that limited supervision and can even fail to benefit from unlabeled data when the labeled set is sufficiently poor. Figure 1c demonstrates the accuracy gain of FreeMatch on the CIFAR-100 dataset—compared to training with labeled data alone—as the volume of unlabeled samples increases under different labeled data scales, where it can be observed that in the one-label-per-class setting, the gain SSL derives from unlabeled samples is substantially smaller than that in other scenarios. This explains the abrupt performance drop in Figure 1a. Therefore, it is crucial to develop mechanisms for utilizing unlabeled data that do not **depend exclusively on** labeled data.

CLIP (Radford et al., 2021) employs contrastive learning to align a large number of image-text pairs, achieving exceptional performance across various vision tasks *without any annotations* (Rao et al., 2022; Shi et al., 2022). Inspired by this, we argue that even when a model trained with restricted labeled data struggles to generate reliable pseudo labels, CLIP's zero-shot capability may act as a catalyst for unlocking the potential of unlabeled data in SSL. However, effectively integrating CLIP into SSL remains challenging. Parameter-efficient fine-tuning (PEFT) methods (Zhou et al., 2022) such as CLIP-Adapter (Gao et al., 2024) enable few-shot adaptation but often fail to capture the diversity present in SSL training data (Figure 2a); conversely, full fine-tuning is prohibitively expensive due to CLIP's fixed input resolution and large parameter count (Gao et al., 2024) (Figure 2b). DebiasPL (Wang et al., 2022a) incorporates CLIP-predicted high-confidence unlabeled samples into the labeled set before training (Figure 2c), but, as Figure 5 and prior work (Wang et al., 2022a) show, CLIP's biased predictions limit the scalability of such approaches.

To address these limitations, we propose **CLIP a**s a **P**rior **T**eacher (**CaPT**), a novel asymmetric-modalities co-training framework that decouples the provision of reliable prior from the provision of strong learning capacity (Figure 2d). Specifically, CaPT jointly trains the multimodal CLIP and a unimodal network, with co-pseudo labels facilitating complementary strengths between the two models. While we fully fine-tune the unimodal network, CLIP is fine-tuned efficiently using PEFT,

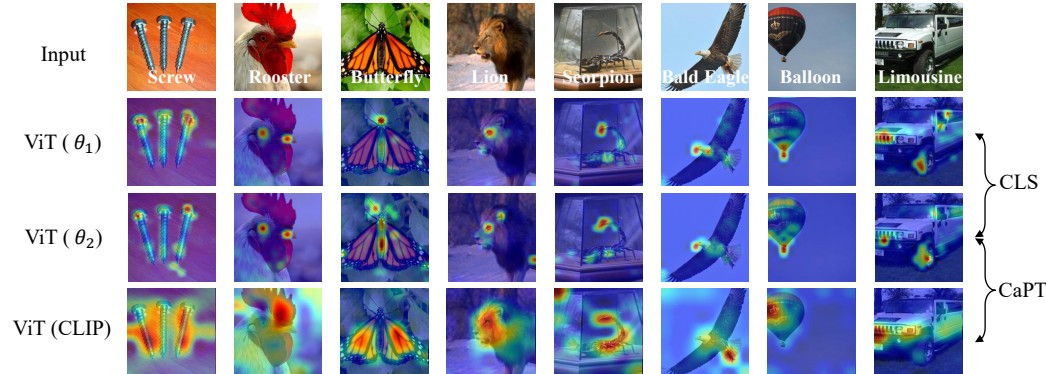

Figure 3: Attention maps for different vision transformers.

where lightweight adapters (Houlsby et al., 2019) are inserted into its textual and visual encoders. The former can better adapt to the richness of the training samples and is not constrained by input image resolution. The latter injects more reliable prior into SSL while maintaining efficiency.

While our emphasis is on efficiently integrating CLIP into SSL, CaPT—like most co-training methods (Yao et al., 2022; Blum & Mitchell, 1998)—preserves bidirectional information flow, enabling mutual learning between models and improving both branches. Crucially, the asymmetric-modalities design mitigates the *pattern-homogeneity bottleneck* encountered when co-training two pure-vision models (e.g., CLS (Yao et al., 2022)). Prior work (Blum & Mitchell, 1998) emphasizes that the independence between cotrained views is essential for successful co-training. However, as shown in Figure 3, unimodal Vision Transformers (ViTs)—despite differing parameter initializations ViT ($\theta_1$) and ViT ($\theta_2$)—still exhibit similar representational patterns. By incorporating textual context, CLIP produces representations that diverge substantially from those of a pure-vision ViT (e.g., on a "rooster" example ViT (CLIP) attends to the comb while the pure-vision ViTs focus on the eye and beak), and this cross-modal complementarity enriches the mutual learning mechanism and markedly enhances co-training effectiveness[1]. The contributions of our work are as follows:

1. We identify and theoretically establish the label dependency that constrains SSL, where the ability to utilize unlabeled data is bounded by the quantity or quality of labeled data.

2. We design a novel and portable framework for integrating CLIP into SSL, CaPT. Its advantages are two-fold: first, it efficiently leverages CLIP's prior knowledge in SSL, unlocking the potential of unlabeled data; second, among co-training methods, it enables richer information exchange between models, enhancing mutual learning.

3. CaPT significantly outperforms existing SSL methods and demonstrates immense potential in realistic restricted supervision scenarios.

## 2 RELATED WORK

In this section, we provide an overview of SSL from two key perspectives: thresholding strategies (Sohn et al., 2020; Zhang et al., 2021a; Wang et al., 2023) and data augmentation techniques (Cubuk et al., 2018; Zhang et al., 2018).

**Thresholding Strategies.** A key direction in SSL research focuses on developing more effective thresholding strategies for generating pseudo labels. Pseudo-labeling (Lee et al., 2013) assigns each unlabeled example the class with highest predicted probability, which is simple but prone to confirmation bias (Arazo et al., 2020). FixMatch (Sohn et al., 2020) applies a fixed confidence threshold to select unlabeled samples for training. Subsequent methods seek greater adaptivity: MPL (Pham et al., 2021) adapts the teacher using student feedback to improve pseudo label quality; Dash (Xu et al., 2021) introduces dynamic filtering based on training loss; FlexMatch (Zhang et al., 2021a) adjusts thresholds for each class according to learning difficulty; FreeMatch (Wang et al.,

---

[1]Due to space constraints, the experiments demonstrating CaPT's cross-modal complementarity are provided in Appendix B.

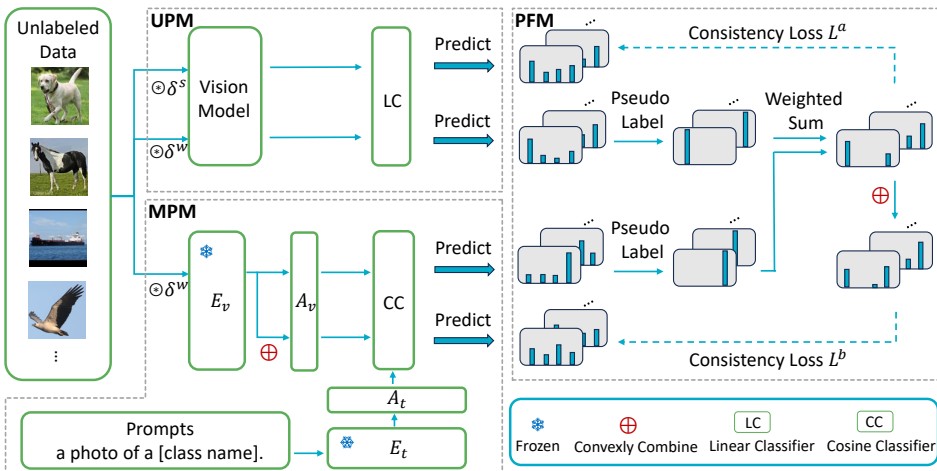

Figure 4: CaPT pipeline: Given a batch of unlabeled images, UPM uses a vision model to extract features and generate predictions from both strongly and weakly augmented views. MPM uses adapter-tuned CLIP and feature augmentation (Mixup) to obtain strong and weak features, and computes cosine similarity with class weights obtained through class prompts for prediction. The weak predictions from both modules are converted into pseudo labels in PFM, which are combined through entropy-based weighting to form co-pseudo labels that supervise strong predictions.

2023) further adapts thresholds to the model's learning state; and SoftMatch (Chen et al., 2023) replaces hard thresholding with confidence-weighted sample contributions.

**Data Augmentation Techniques.** Data augmentation is another central pillar of modern SSL. VAT (Miyato et al., 2018) enforces consistency between original and adversarially perturbed inputs to improve robustness. Mixup (Zhang et al., 2018) augments training by convexly combining pairs of samples and has been shown to effectively smooth decision boundaries in SSL (Berthelot et al., 2019b;a; Han et al., 2025). FixMatch and follow-up works (Li et al., 2021; Nguyen, 2024) refine augmentation schemes by integrating stronger strategies such as RandAugment (Cubuk et al., 2020), which improves generalization. FlatMatch (Huang et al., 2023) applies sharpness-aware minimization (Foret et al., 2020) to flatten the loss landscape and further enhance generalization.

However, as shown in Figure 1a, existing SSL algorithms exhibit suboptimal performance under weak supervision (see Appendix J for further scenarios). We refine CLIP's prior and integrate it into SSL, effectively reducing SSL's label dependency. Our work is related to DebiasPL (Wang et al., 2022a) and CLS (Yao et al., 2022), which incorporate CLIP and co-training into SSL, respectively. Unlike CLS, which co-trains two unimodal networks with identical architectures but different parameter initializations, CaPT jointly trains models from asymmetric-modalities, breaking SSL's label dependency and enabling informative co-training. Compared to DebiasPL, we utilize CLIP in a more reliable manner and identify its critical role in restricted supervision scenarios.

## 3 METHOD

As illustrated in Figure 4, the overall workflow of CaPT is structured into three modules: **Unimodal Prediction Module (UPM)**, where a unimodal network generates predictions for unlabeled samples; **Multimodal Prediction Module (MPM)**, where CLIP is fine-tuned and used to produce predictions for the same samples; and **Prediction Fusion Module (PFM)**, which aggregates the predictions from the first two modules and computes the loss.

### 3.1 UNIMODAL PREDICTION MODULE

The process in UPM follows common practices (Sohn et al., 2020) in current SSL methods. Given an unlabeled sample $x_u$, we apply weak and strong augmentations to it, and then obtain the predictions

for both augmented views using the classification model:

$$q^{w,a} = p_m(y|x_u \circledast \delta^w), q^{s,a} = p_m(y|x_u \circledast \delta^s). \tag{2}$$

Here, we use the symbol $\circledast$ to denote the augmentation operation applied to data, $\delta^w$ and $\delta^s$ represent weak and strong augmentations, respectively, $q^{w,a}$ ($q^{s,a}$) denotes the weak (strong) prediction generated in UPM, and $p_m(y|x)$ is the predicted class distribution produced by the model for input $x$.

To better illustrate our algorithm, we present the subsequent usual practice in SSL: converting the weakly augmented view's prediction into pseudo label:

$$\hat{q} = \arg\max(q^{w,a}), \tag{3}$$

then calculating the consistency loss between the prediction of the strongly augmented sample and the pseudo label:

$$l = CE(\hat{q}, q^{s,a}), \tag{4}$$

where $CE(\cdot, \cdot)$ denotes the standard cross-entropy loss. The core idea is consistency regularization (Bachman et al., 2014): expecting the model to maintain consistent predictions before and after perturbations to enhance generalization ability.

Our method converts the weak predictions from UPM into pseudo labels. To integrate CLIP's prior knowledge, we combine them with the pseudo labels generated in MPM to form co-pseudo labels, which serve as the supervision signal for strong predictions.

## 3.2 MULTIMODAL PREDICTION MODULE

MPM extracts reliable prior knowledge from CLIP. We find that fully fine-tuning CLIP and obtaining predictions for both weak and strong views of input data through it is much more time-consuming compared to UPM. To mitigate this, we introduce adapter for efficient fine-tuning of CLIP and employ feature-augmented consistency regularization.

### 3.2.1 ADAPTER-TUNING

CLIP typically requires input images with resolutions of $224 \times 224$ or higher, and its encoders contain a large number of parameters, prompting us to explore an efficient fine-tuning method. We freeze the visual and textual encoders of CLIP and only train additional adapters, as commonly done in few-shot learning (Gao et al., 2024).

Specifically, given a weakly augmented version of an unlabeled sample $x_u \circledast \delta^w$, we first extract features through CLIP's visual encoder $E_v$:

$$f = E_v(x_u \circledast \delta^w), \tag{5}$$

and then input $f$ into a learnable adapter $A_v$, which consists of two linear layers for dimensionality reduction and expansion. The output is then combined with the original feature $f$ using a residual connection to obtain the fine-tuned feature:

$$f^* = f + A_v(f). \tag{6}$$

To construct class weights for classification, we follow the zero-shot CLIP, placing each class name into a predefined template to generate class prompts (more details in Appendix H), which are then input into CLIP's textual encoder $E_t$ to obtain class weights $W$. We simplify the fine-tuning of the textual encoder by constructing a learnable parameter $A_t$ (Zhu et al., 2023), initialized to zero and of the same shape as $W$, and similarly combining it with class weights $W$ using a residual connection to obtain new class weights:

$$W^* = W + A_t. \tag{7}$$

With the fine-tuned image feature $f^*$ and class weights $W^*$, we can use cosine similarity as a classifier to calculate the predicted probability of the sample for each class:

$$p_i = \frac{\exp\left(\frac{W_i^{*T} f^*}{\tau}\right)}{\sum_{j=1}^{C} \exp\left(\frac{W_j^{*T} f^*}{\tau}\right)}, \tag{8}$$

where $\tau$ denotes the temperature of softmax, $W_i^*$ represents the prototype weight vector for class $i$.

### 3.2.2 Feature-Augmented Consistency Regularization

Given than CLIP's visual encoder $E_v$ is frozen, we implement strong augmentation at feature level instead of input level to reduce resource consumption. Mixup (Zhang et al., 2018) achieves data augmentation through convex combinations of data and labels, and feature-based Mixup has shown strong potential (Verma et al., 2019). Inspired by this, we perform feature-level strong augmentation by convexly combining the feature $f$ extracted from weakly augmented anchor sample by $E_v$ with the feature $f'$ of another weakly augmented sample randomly selected from the same batch:

$$\bar{f} = \lambda f + (1 - \lambda)f', \tag{9}$$

where $\lambda$ follows a Beta distribution with parameters $(\alpha, \alpha)$. Next, we fine-tune the feature $f$ and $\bar{f}$ using the adapter $A_v$, obtaining $f^*$ and $\bar{f}^*$, respectively. We then use the fine-tuned features and class weights $W^*$ to obtain the final weak and strong predictions $q^{w,b}$ and $q^{s,b}$ via Equation 8. The strong prediction $q^{s,b}$ will later be paired with a convexly combined co-pseudo label to enforce feature-augmented consistency regularization in PFM.

Feature-augmented consistency regularization not only improves the generalization of CLIP but also avoids the need to construct another high-resolution version of the unlabeled image and feed it again to the parameter-heavy visual encoder $E_v$ to obtain feature.

### 3.3 Prediction Fusion Module

Modules UPM and MPM generate strong and weak predictions using the unimodal network and CLIP, respectively. To aggregate the two modules for exchanging supervision signals, we weight and combine the pseudo labels generated by these modules to form co-pseudo labels. Specifically, we first construct pseudo labels from the weak predictions of the two modules:

$$\hat{q}^a = \arg\max(q^{w,a}), \hat{q}^b = \arg\max(q^{w,b}). \tag{10}$$

To assign weights, given a batch of unlabeled samples $x_j^u$, where $j \in (1 \ldots N)$, assume any model's weakly augmented version predictions for these samples are $q_j^w$. We first compute the average entropy of the model's predictions for these samples:

$$H = \frac{1}{N} \sum_{j=1}^{N} \left( -\sum_i q_{j,i}^w \log q_{j,i}^w \right), \tag{11}$$

where $q_{j,i}^w$ represents the predicted probability for class $i$ of the $j$-th sample. The smaller the entropy, the higher the model's confidence in its predictions, which warrants a higher weight allocation. The weights assigned to the two modules are defined as:

$$\Gamma^a = \frac{\frac{1}{H^a}}{\frac{1}{H^a} + \frac{1}{H^b}}, \Gamma^b = \frac{\frac{1}{H^b}}{\frac{1}{H^a} + \frac{1}{H^b}}. \tag{12}$$

Then we combine the two pseudo labels using the computed weights to generate a co-pseudo label:

$$\hat{q}^c = \Gamma^a \hat{q}^a + \Gamma^b \hat{q}^b. \tag{13}$$

Entropy-based weighting enables adaptive weight adjustment. At the early co-training, the unimodal network is not yet fully trained, while CLIP, with its rich prior knowledge, dominates the supervision. As the full fine-tuning of the network in UPM progresses, it gradually takes over the supervision due to its larger number of learnable parameters.

To parallel the feature level convex combination in MPM, the co-pseudo label of the anchor sample is convexly combined with that of the randomly selected sample using the same mixing parameter $\lambda$:

$$\bar{\hat{q}}^c = \lambda \hat{q}^c + (1 - \lambda)\hat{q}^{c'}. \tag{14}$$

Ultimately, the strong predictions from both modules jointly use the co-pseudo label (or its mixed variant) as the supervision signal to compute the consistency loss:

$$L^a = \frac{1}{N} \sum_{j=1}^{N} CE(\hat{q}_j^c, q_j^{s,a}), L^b = \frac{1}{N} \sum_{j=1}^{N} CE(\bar{\hat{q}}_j^c, q_j^{s,b}). \tag{15}$$

Table 1: Accuracy (%) on CIFAR-100, STL10 and EuroSAT datasets under USB. The best results are highlighted with **Bold** and the second-best results are highlighted with underline.

| Dataset | CIFAR-100 | | STL10 | | EuroSAT | |
|---|---|---|---|---|---|---|
| # Labels per Class | 2 | 4 | 4 | 10 | 2 | 4 |
| VAT (Miyato et al., 2018) | $68.51_{\pm1.33}$ | $78.66_{\pm0.50}$ | $81.55_{\pm1.47}$ | $89.31_{\pm0.51}$ | $73.84_{\pm0.96}$ | $89.91_{\pm0.94}$ |
| Mean Teacher (Tarvainen & Valpola, 2017) | $64.53_{\pm0.40}$ | $73.97_{\pm0.30}$ | $81.33_{\pm1.69}$ | $75.81_{\pm10.15}$ | $73.17_{\pm1.46}$ | $84.15_{\pm1.66}$ |
| ReMixMatch (Berthelot et al., 2019a) | $77.79_{\pm2.21}$ | $83.14_{\pm0.57}$ | $86.92_{\pm3.34}$ | $92.79_{\pm0.39}$ | $94.95_{\pm1.05}$ | $94.93_{\pm0.56}$ |
| FixMatch (Sohn et al., 2020) | $70.40_{\pm0.90}$ | $80.44_{\pm0.52}$ | $83.85_{\pm1.89}$ | $91.89_{\pm0.68}$ | $86.56_{\pm3.53}$ | $94.09_{\pm2.02}$ |
| FlexMatch (Zhang et al., 2021a) | $73.24_{\pm1.12}$ | $81.76_{\pm0.36}$ | $85.60_{\pm3.11}$ | $91.83_{\pm0.78}$ | $94.83_{\pm0.57}$ | $94.42_{\pm0.81}$ |
| Dash (Xie et al., 2020) | $69.39_{\pm0.98}$ | $80.62_{\pm0.10}$ | $83.78_{\pm5.95}$ | $92.15_{\pm0.74}$ | $88.81_{\pm0.90}$ | $93.04_{\pm0.87}$ |
| CoMatch (Li et al., 2021) | $64.92_{\pm0.69}$ | $74.77_{\pm0.50}$ | $84.88_{\pm1.88}$ | $90.44_{\pm1.35}$ | $94.25_{\pm0.43}$ | $95.19_{\pm1.05}$ |
| SimMatch (Zheng et al., 2022) | $76.22_{\pm1.08}$ | $82.94_{\pm0.78}$ | $88.23_{\pm3.20}$ | $92.45_{\pm1.86}$ | $92.34_{\pm0.60}$ | $94.73_{\pm0.89}$ |
| SoftMatch (Chen et al., 2023) | $77.33_{\pm1.32}$ | $83.16_{\pm0.66}$ | $86.45_{\pm3.16}$ | $92.16_{\pm1.72}$ | $94.25_{\pm0.62}$ | $94.10_{\pm1.42}$ |
| FreeMatch (Wang et al., 2023) | $78.60_{\pm0.30}$ | $84.35_{\pm0.26}$ | $87.27_{\pm3.22}$ | $91.48_{\pm0.53}$ | $93.50_{\pm0.78}$ | $94.22_{\pm0.51}$ |
| SequenceMatch (Nguyen, 2024) | $78.86_{\pm0.25}$ | $84.26_{\pm0.15}$ | $87.73_{\pm3.13}$ | $92.45_{\pm0.66}$ | $94.13_{\pm0.69}$ | $95.09_{\pm0.81}$ |
| RegMixMatch (Han et al., 2025) | $\underline{80.74}_{\pm0.56}$ | $\underline{84.45}_{\pm0.31}$ | $\underline{89.89}_{\pm3.20}$ | $\underline{92.90}_{\pm0.66}$ | $\underline{95.75}_{\pm0.77}$ | $\underline{96.39}_{\pm0.75}$ |
| **CaPT** | $\mathbf{84.83}_{\pm0.10}$ | $\mathbf{85.60}_{\pm0.07}$ | $\mathbf{96.07}_{\pm0.05}$ | $\mathbf{96.34}_{\pm0.05}$ | $\mathbf{96.60}_{\pm0.13}$ | $\mathbf{96.98}_{\pm0.11}$ |
| Adapter-tuned CLIP | $74.90_{\pm0.03}$ | $75.54_{\pm0.02}$ | $96.86_{\pm0.01}$ | $97.15_{\pm0.01}$ | $93.83_{\pm0.06}$ | $94.52_{\pm0.04}$ |
| CLIP (Radford et al., 2021) | 65.10 | | 97.18 | | 49.46 | |

We emphasize that a pseudo label is retained only if the weak-prediction confidence exceeds a threshold. Otherwise, the corresponding module's pseudo label is replaced by the all-zero vector. Consequently, the resulting co-pseudo label comprises supervision exclusively from the other module, with the sum of class prediction probabilities equaling the module's weight (less than 1). This mechanism ensures that low-confidence samples are assigned smaller weights during the loss calculation, thus reducing confirmation bias.

Overall, CaPT aggregates the weak predictions of the unimodal network and the multimodal CLIP through co-pseudo labels, guiding the strong predictions to align with these co-pseudo labels to jointly train the unimodal network and fine-tune CLIP. Although CLIP is only adapter-tuned and thus typically lags behind the fully fine-tuned network in UPM, it remains indispensable, as its adapter-tuned outputs provide reliable priors that serve as a catalyst for effectively leveraging unlabeled samples in SSL.

## 4 EXPERIMENTS

This section provides a comprehensive experimental evaluation of CaPT. We begin by reporting its performance on the USB benchmark (Wang et al., 2022b). Next, we evaluate CaPT on large-scale datasets, extreme low-label settings, and fine-grained benchmarks. Finally, we conduct extensive ablation studies to validate the design choices of CaPT.

### 4.1 USB

USB (Wang et al., 2022b) is a unified SSL benchmark for the fair evaluation of SSL methods. It adopts the pre-trained ViTs (Dosovitskiy, 2020) as the training backbone. For fair comparison, our unimodal network uses the same training configuration and backbone as USB. Unless otherwise stated, ViT-B/32 is employed as the visual encoder for CLIP. Detailed experimental configurations are provided in Appendix F. Similar to RegMixMatch, we validate CaPT on three datasets, including CIFAR-100 (Krizhevsky, 2009), STL-10 (Coates et al., 2011), and EuroSAT (Helber et al., 2019), under different amounts of labeled data conditions. The experimental results are compared against 12 established SSL algorithms, including VAT (Miyato et al., 2018), Mean-Teacher (Tarvainen & Valpola, 2017), ReMixMatch (Berthelot et al., 2019a), FixMatch (Sohn et al., 2020), FlexMatch (Zhang et al., 2021a), Dash (Xu et al., 2021), CoMatch (Li et al., 2021), Sim-Match (Zheng et al., 2022), FreeMatch (Wang et al., 2023), SoftMatch (Chen et al., 2023), SequenceMatch (Nguyen, 2024), and RegMixMatch (Han et al., 2025). Each algorithm is trained three times with different random seeds. We adopt the adaptive threshold strategy from FreeMatch to filter pseudo labels, as in RegMixMatch. The final performance of CaPT is reported using the fully fine-tuned unimodal network, while the results of our adapter-tuned CLIP are also presented.

Table 1 presents the results of CaPT on the USB benchmark[2]. The results show that CaPT leads in all 6 commonly used evaluation settings, with particularly significant improvements when labeled data is scarce. Specifically, with only 2 labeled samples per class on the CIFAR-100 dataset, CaPT outperforms the second-best method (i.e., RegMixMatch) by 4.09%. On the STL-10 dataset, where each class has 4 labeled samples, CaPT leads by 6.18%. Moreover, under varying labeled data quality (i.e., different random seeds), CaPT achieves a lower standard deviation compared to existing SSL methods. These results illustrate that CaPT effectively mitigates SSL's label dependency on labeled data.

Table 2: Performances of CaPT on the ImageNet dataset.

| Dataset | ImageNet | | | |
|---|---|---|---|---|
| # Labels per Class | 10 | | 100 | |
| Top-$n$ acc | Top-1 | Top-5 | Top-1 | Top-5 |
| FixMatch | 53.61 | 75.96 | 71.53 | 90.36 |
| FlexMatch | 54.21 | 76.80 | 72.17 | 90.59 |
| FreeMatch | 54.69 | 77.02 | 72.57 | 90.97 |
| RegMixMatch | 58.35 | 81.10 | 73.66 | 91.89 |
| **CaPT** | **67.68** | **89.82** | **74.21** | **92.19** |

## 4.2 IMAGENET

To verify the scalability of CaPT, we conduct experiments on the ImageNet dataset (Deng et al., 2009). Similar to RegMixMatch, we use MAE pre-trained ViT-B (He et al., 2022) as the training backbone for UPM and conduct experiments with 10 and 100 labeled samples per class. Table 2 demonstrates CaPT's superior performance on the ImageNet dataset. The leading advantage of CaPT is more pronounced under conditions with fewer labels: with 10 labeled samples per class, CaPT outperforms RegMixMatch by 9.33%.

## 4.3 EXTREMELY-SCARCE-LABELS REGIMES

To further assess CaPT under extreme label scarcity, we evaluate on CIFAR-10, CIFAR-100 and EuroSAT with only one labeled sample per class, and compare against state-of-the-art SSL methods FreeMatch and RegMixMatch (Table 3). CaPT attains a substantial lead in the one-label-per-class setting. Notably, when labeled samples per class decrease from 2 to 1, FreeMatch and RegMixMatch suffer accuracy drops of 17.47% (78.60%→61.13%) and 20.25% (80.74%→60.49%) on CIFAR-100, respectively. By contrast, CaPT substantially reduces reliance on labeled data: in the one-label-per-class setting it improves over the second-best method by 21.38% on CIFAR-100 and by 4.05% on EuroSAT[3]. These results indicate that, although recent SSL algorithms perform well with moderate labels, their accuracy degrades sharply under extreme scarcity, whereas CaPT remains robust by refining and leveraging CLIP's prior, effectively unlocking the potential of unlabeled data.

Table 3: Performances of different SSL methods with one labeled sample per class.

| Dataset | CIFAR-10 | CIFAR-100 | EuroSAT |
|---|---|---|---|
| FreeMatch | 91.93 | 61.13 | 90.12 |
| RegMixMatch | 95.65 | 60.49 | 92.28 |
| **CaPT** | **96.37** | **82.51** | **96.33** |

Furthermore, in Table 4, we compare the training time and memory consumption of CaPT with other SSL algorithms. The experiments are conducted on a 10GB RTX 3080 GPU. The results show that, compared to training a unimodal network (FreeMatch), CaPT achieves significant performance improvement at the cost of only 8.00% more memory consumption and 11.18% additional training time. Compared to the latest state-of-the-art method RegMixMatch, CaPT demonstrates significant advantages in both resource consumption and classification performance.

Table 4: Comparison of training time and memory consumption on CIFAR-100 with 2 labeled samples per class.

| Method | Time (sec./iter.) | Mem. (MiB) | Acc. (%) |
|---|---|---|---|
| FreeMatch | 0.0939 | 4676 | 78.60 |
| RegMixMatch | 0.1484 | 6578 | 80.74 |
| CaPT | 0.1044 | 5050 | 84.83 |

---

[2]The results of CaPT using TorchSSL (Zhang et al., 2021a), along with broader method comparisons, are provided in Appendix I.

[3]We note that under the 1-shot setting, FreeMatch already achieves near-saturation on EuroSAT (90.12%). A 6.21% improvement by CaPT (96.33%) on such a strong baseline is therefore also substantial.

Table 5: Accuracy (%) of CaPT on fine-grained datasets.

| Dataset | FGVCAircraft | | Flowers102 | | StanfordCars | | SUN397 | | DTD | | SVHN | |
|---|---|---|---|---|---|---|---|---|---|---|---|---|
| # Labels per Class | 5 | 10 | 1 | 2 | 5 | 10 | 30 | 50 | 10 | 20 | 2 | 4 |
| FreeMatch | **51.43** | 65.82 | 75.31 | 93.89 | 66.31 | 83.79 | 70.06 | 74.87 | 62.14 | 70.03 | 67.35 | 86.61 |
| RegMixMatch | 49.86 | **66.21** | 80.23 | 94.31 | 68.75 | 85.60 | 72.57 | 76.59 | 63.56 | 71.06 | 70.23 | 76.56 |
| **CaPT** | 50.12 | 64.33 | **94.71** | **96.42** | **80.36** | **89.66** | **75.69** | **78.29** | **66.22** | **72.93** | **81.20** | **91.73** |
| CLIP | 18.97 | | 61.33 | | 52.63 | | 59.53 | | 40.17 | | 34.36 | |

## 4.4 FINE-GRAINED DATASETS

To preclude any advantage for CaPT arising from potential overlap between CLIP's corpus and simple benchmarks[4], Table 5 presents its performance on 6 fine-grained benchmarks: FGVCAircraft (Maji et al., 2013), Flowers102 (Nilsback & Zisserman, 2008), StanfordCars (Krause et al., 2013), SUN397 (Xiao et al., 2010), DTD (Cimpoi et al., 2014), and SVHN (Netzer et al., 2011). Except for FGVCAircraft (discussed in Appendix N), CaPT outperforms competing methods across all other datasets, underscoring its great scalability on datasets with greater domain shift.

## 4.5 ABLATION STUDY

In this section, we validate the design choices behind CaPT. We begin by validating the effectiveness of the CaPT framework in leveraging CLIP within SSL. We evaluate three ablated variants of CaPT: CaPT-Ada (Figure 2a), in which the model in UPM is replaced with a CLIP-Adapter and MPM is removed; CaPT-Deb (Figure 2c), which disables adapter-tuning and vision model→CLIP flow; and CaPT-Uni (Figure 2d but retaining only the CLIP→vision model flow), where the unimodal network no longer transmits information back to CLIP. As reported in Table 6, the full CaPT achieves the best performance across all settings. Despite access to abundant data, CaPT-Ada suffers a substantial performance drop (-16.40% and -16.38%) due to the lack of sufficient learnable parameters. CaPT-Deb, affected by CLIP's biased prior, shows a significant decline (-12.73%) on the EuroSAT dataset, which is more sensitive to CLIP's class preference. This highlights the importance of adapter tuning in mitigating CLIP's biased prior (see Figure 5). Lastly, the performance degradation of CaPT-Uni confirms that maintaining bidirectional knowledge exchange between models is crucial for enhancing overall effectiveness.

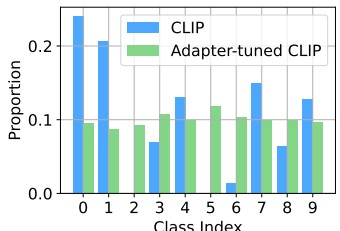

Figure 5: Adapter-tuning effectively mitigates CLIP's biased prior. Experiments are conducted on EuroSAT datasets with 2 labeled samples per class.

Additionally, we show the contribution of unimodal network (i.e., only UPM) and multimodal CLIP (i.e., only MPM) during co-training by retaining only one of the two classification models. Retaining only the unimodal network results in degraded performance (-6.23% and -3.10%), as the model lacks the prior knowledge needed to fully exploit unlabeled samples. Conversely, when only MPM is retained, although CLIP provides prior knowledge, the limited number of learnable parameters cannot match the richness of the unlabeled samples, ultimately leading to a significant performance decline (-16.51% and -16.48%) due to underfitting. The co-training mechanism proposed in CaPT effectively combines the strength of these two modules.

Table 6: Ablation study of CaPT. Experiments are conducted with 2 labeled samples per class.

| Dataset | CIFAR-100 | EuroSAT |
|---|---|---|
| CaPT | **84.83** | **96.60** |
| CaPT-Ada | 68.43 (-16.40) | 80.22 (-16.38) |
| CaPT-Deb | 81.03 (-3.80) | 83.87 (-12.73) |
| CaPT-Uni | 83.95 (-0.88) | 95.11 (-1.49) |
| only UPM | 78.60 (-6.23) | 93.50 (-3.10) |
| only MPM | 68.32 (-16.51) | 80.12 (-16.48) |
| w/o feat aug. | 84.26 (-0.57) | 94.79 (-1.81) |
| equal weights | 83.96 (-0.87) | 95.03 (-1.57) |

Finally, we evaluate the impact of feature-augmented consistency regularization and entropy-based weighting. Removing consistency regularization on the CLIP side (i.e., w/o feat aug.) leads to performance degradation, indicating that improving the generalization of either branch benefits the overall system. Replacing entropy-based weighting with equal weighting (i.e., 0.5 per model) also

---

[4]We offer an in-depth discussion of this issue in Appendix M.

results in a performance drop, suggesting that entropy-based weighting more effectively adapts to the training dynamics of each model, thereby enhancing final performance.

## 5 Conclusion, Limitation and Broader Impact

In this paper, we identify and theoretically demonstrate an inherent limitation of SSL: deficiencies in labeled data undermine the accuracy of pseudo labels, thereby hindering the effective use of unlabeled data. We propose CaPT, a novel framework that integrates vision-language models (VLMs) into SSL, efficiently and reliably leveraging CLIP to mitigate the label dependency of SSL. CaPT exhibits exceptional performance and efficiency across public benchmarks, extremely-scarce-labels regimes, fine-grained datasets and more realistic weakly supervised SSL settings (Appendix J).

Despite its overall effectiveness, we observe that CLIP's prior is less informative on certain fine-grained datasets such as FGVCAircraft, limiting its contribution in this case. Nonetheless, CaPT's primary contribution lies in **establishing a general and future-proof framework for integrating VLMs into SSL**. As more powerful VLMs (Sun et al., 2023) emerge, they can be seamlessly[5] incorporated into CaPT to efficiently achieve strong performance (see Appendix N). Developing more efficient frameworks for integrating VLMs into SSL could be a potential future direction.

## Reproducibility Statement

We provide detailed descriptions of our model architecture, training procedure, and hyperparameter settings in Section 3, Section 4, and Appendix F. Complete proofs of Theorem 1.1 are included in Appendix A. Anonymous source code for reproduction is provided in the supplementary materials.

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

# A  PROOF OF THEOREM 1.1 (NEAREST-PROTOTYPE PSEUDO LABEL ERROR BOUND)

**Notation and model assumptions.**  Let $(x, y) \sim \mathcal{D}$ with $y \in \{1, \ldots, K\}$. For each class $c$ assume the conditional distribution of $x$ is

$$x \mid y = c \sim \mathcal{N}(\mu_c, \sigma^2 I_d),$$

and define the minimum inter-class centroid distance

$$g := \min_{c \neq c'} \|\mu_c - \mu_{c'}\| > 0.$$

For each class $c$ we are given $n_c \geq 1$ labeled samples and form the sample mean (prototype)

$$m_c := \frac{1}{n_c} \sum_{i:y_i=c} x_i.$$

Decompose

$$m_c = \mu_c + b_c + \delta_c,$$

where $b_c$ is a deterministic bias vector (systematic selection bias / non-prototypicality), and

$$\delta_c \sim \mathcal{N}\left(0, \frac{\sigma^2}{n_c} I_d\right)$$

is the random sampling error of the sample mean. Assume the deterministic bias satisfies $\|b_c\| \leq B$ for all $c$.

We adopt the nearest-prototype decision rule for unlabeled $x$:

$$\hat{y}(x) = \arg\min_c \|x - m_c\|.$$

Below we will prove the following high-probability bound.

**Theorem A.1** (Theorem 1.1). *Let $n_{\min} := \min_c n_c$. Fix $\eta \in (0, 1)$ and define*

$$\varepsilon_n := \frac{2\sigma}{\sqrt{n_{\min}}} \sqrt{\log\left(\frac{K \, 2^{d/2}}{\eta}\right)}, \qquad r := B + \varepsilon_n.$$

*Assume $g/2 > r$ (otherwise the bound below is vacuous). Then with probability at least $1 - \eta$ (over the randomness of the labeled samples) the following holds for every class $c$:*

$$\Pr_{x \sim \mathcal{N}(\mu_c, \sigma^2 I_d)} \left(\hat{y}(x) \neq c\right) \leq (K - 1) \, 2^{d/2} \exp\left(-\frac{(g/2 - r)^2}{4\sigma^2}\right). \tag{16}$$

**Lemma A.2.** *Let $Z \sim \mathcal{N}(0, I_d)$ and $W = \|Z\|^2$. Then for any $u > 0$ and any $\lambda \in (0, 1/2)$,*

$$\Pr(W \geq u) \leq (1 - 2\lambda)^{-d/2} \exp(-\lambda u).$$

*In particular, choosing $\lambda = \frac{1}{4}$ yields*

$$\Pr\left(\|Z\| \geq t\right) = \Pr\left(W \geq t^2\right) \leq 2^{d/2} \exp\left(-\frac{t^2}{4}\right), \qquad (t > 0). \tag{17}$$

*Proof.* The mgf of a $\chi_d^2$ variable $W$ satisfies $\mathbb{E}[e^{\lambda W}] = (1 - 2\lambda)^{-d/2}$ for $\lambda < 1/2$. By Markov (Chernoff) bound,

$$\Pr(W \geq u) \leq \mathbb{E}[e^{\lambda W}] e^{-\lambda u} = (1 - 2\lambda)^{-d/2} e^{-\lambda u}.$$

Setting $\lambda = \frac{1}{4}$ gives $(1 - 2\lambda)^{-d/2} = (1/2)^{-d/2} = 2^{d/2}$ and the stated bound follows by taking $u = t^2$.

$\square$

**Lemma A.3.** *For each class $c$, $\delta_c \sim \mathcal{N}(0, \sigma^2/n_c\, I_d)$. Hence for any $t > 0$,*

$$\Pr\left(\|\delta_c\| \geq t\right) \leq 2^{d/2} \exp\left(-\frac{n_c t^2}{4\sigma^2}\right). \tag{18}$$

*Proof.* Let $Z_c := \sqrt{\frac{n_c}{\sigma^2}}\, \delta_c \sim \mathcal{N}(0, I_d)$. Applying Lemma A.2 with $t' = \sqrt{\frac{n_c}{\sigma^2}}\, t$ yields

$$\Pr(\|\delta_c\| \geq t) = \Pr\left(\|Z_c\| \geq t'\right) \leq 2^{d/2} \exp\left(-\frac{(t')^2}{4}\right) = 2^{d/2} \exp\left(-\frac{n_c t^2}{4\sigma^2}\right),$$

which is precisely Equation 18.

$\square$

*Proof of Theorem A.1.* Let $n_{\min} = \min_c n_c$. Fix $\eta \in (0, 1)$ and choose $\varepsilon_n > 0$ such that

$$K\, 2^{d/2} \exp\left(-\frac{n_{\min}\varepsilon_n^2}{4\sigma^2}\right) \leq \eta. \tag{19}$$

Solving Equation 19 for $\varepsilon_n$ gives the explicit choice

$$\varepsilon_n = \frac{2\sigma}{\sqrt{n_{\min}}} \sqrt{\log\left(\frac{K\, 2^{d/2}}{\eta}\right)}.$$

By Lemma A.3 and a union bound over the $K$ classes,

$$\Pr\left(\exists c:\ \|\delta_c\| \geq \varepsilon_n\right) \leq \sum_{c=1}^{K} \Pr(\|\delta_c\| \geq \varepsilon_n) \leq K\, 2^{d/2} \exp\left(-\frac{n_{\min}\varepsilon_n^2}{4\sigma^2}\right) \leq \eta.$$

Thus with probability at least $1 - \eta$ (over the labeled-sample draw) the following event $\mathcal{E}$ holds:

$$\mathcal{E}:\quad \forall c,\ \|\delta_c\| \leq \varepsilon_n.$$

Condition on $\mathcal{E}$. Then for every class $c$ we have

$$\|m_c - \mu_c\| = \|b_c + \delta_c\| \leq \|b_c\| + \|\delta_c\| \leq B + \varepsilon_n = r. \tag{20}$$

Fix a class $c$ and consider an unlabeled example $x$ with true label $y = c$. Suppose $x$ is misclassified by the nearest-prototype rule, i.e. $\hat{y}(x) \neq c$. Then there exists some $c' \neq c$ such that

$$\|x - m_{c'}\| \leq \|x - m_c\|.$$

For a fixed wrong class $c' \neq c$, if $\hat{y}(x) = c'$ (and $y = c$) then the above inequality holds. Using the triangle inequality and Equation 20 we obtain

$$\|x - m_{c'}\| \geq \|\mu_{c'} - \mu_c\| - \|x - \mu_c\| - \|m_{c'} - \mu_{c'}\| \geq g - \|x - \mu_c\| - r,$$
$$\|x - m_c\| \leq \|x - \mu_c\| + \|m_c - \mu_c\| \leq \|x - \mu_c\| + r.$$

Combining these two inequalities and using $\|x - m_{c'}\| \leq \|x - m_c\|$ gives

$$g - \|x - \mu_c\| - r \leq \|x - m_{c'}\| \leq \|x - m_c\| \leq \|x - \mu_c\| + r,$$

which rearranges to

$$2\|x - \mu_c\| \geq g - 2r \quad \Longrightarrow \quad \|x - \mu_c\| \geq \frac{g}{2} - r.$$

Thus for this fixed $c'$ we have the set inclusion

$$\{\hat{y}(x) = c'\} \subseteq \left\{\|x - \mu_c\| \geq \frac{g}{2} - r\right\},$$

and hence

$$\Pr\left(\hat{y}(x) = c' \mid \mathcal{E}\right) \leq \Pr\left(\|x - \mu_c\| \geq \frac{g}{2} - r\right).$$

By the union bound we obtain

$$\Pr\left(\hat{y}(x) \neq c \mid \mathcal{E}\right) = \Pr\left(\bigcup_{c' \neq c} \{\hat{y}(x) = c'\} \,\Big|\, \mathcal{E}\right) \leq \sum_{c' \neq c} \Pr\left(\hat{y}(x) = c' \mid \mathcal{E}\right) \leq (K-1) \Pr\left(\|x-\mu_c\| \geq \frac{g}{2}-r\right).$$
(21)

Applying Lemma A.2 to the centered Gaussian $(x - \mu_c)/\sigma \sim \mathcal{N}(0, I_d)$ with $t := (g/2 - r)/\sigma > 0$ yields

$$\Pr\left(\|x - \mu_c\| \geq \frac{g}{2} - r\right) \leq 2^{d/2} \exp\left(-\frac{(g/2 - r)^2}{4\sigma^2}\right).$$

Combining this with Equation 21 gives, conditioned on $\mathcal{E}$,

$$\Pr\left(\hat{y}(x) \neq c \mid \mathcal{E}\right) \leq (K - 1)\, 2^{d/2} \exp\left(-\frac{(g/2 - r)^2}{4\sigma^2}\right).$$

Since $\mathcal{E}$ holds with probability at least $1 - \eta$ (over labeled-sample randomness), we conclude that with probability at least $1 - \eta$ (over the labeled-sample draw) the unconditional misclassification probability for class $c$ is upper-bounded by the right-hand side above. This proves Equation 16.

$\square$

## B COMPLEMENTARITY OF CAPT'S ASYMMETRIC-MODALITIES CO-TRAINING

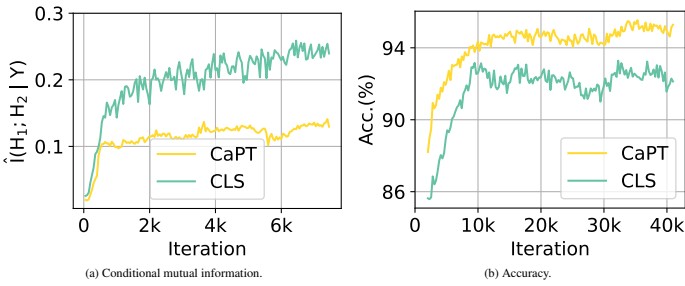

(a) Conditional mutual information.  (b) Accuracy.

Figure 6: Comparison of conditional mutual information and accuracy between CaPT and CLS. Experiment is conducted on the EuroSAT (Helber et al., 2019) dataset with 2 labeled samples per class.

Table 7: Comparison of accuracy (%) between CaPT and CLS.

| Dataset | CIFAR-100 | | EuroSAT | |
|---|---|---|---|---|
| # Labels per Class | 2 | 4 | 2 | 4 |
| CLS | 80.64 | 84.66 | 94.46 | 94.89 |
| **CaPT** | **84.83** | **85.60** | **96.60** | **96.98** |

We perform an in-depth analysis of the complementarity conferred by CaPT's asymmetric-modalities co-training framework. First, if the two co-trained networks consistently produce identical predictions on the same unlabeled batch, co-training reduces to standard single-model training. Second, Blum and Mitchell (Blum & Mitchell, 1998) theoretically proved that if, for each instance, the predictions obtained from the two views $X_1$ and $X_2$ approach conditional independence given the true label $Y$, then the co-training algorithm is PAC-learnable.

Building on these insights, we reasonably hypothesize that, under the CaPT framework, the two co-trained models' predictions on the same sample

$$H_1 = h_1(X), \quad H_2 = h_2(X)$$
(22)

are **closer to conditional independence** given $Y$ than those produced by the symmetric modality co-training baseline (CLS (Yao et al., 2022)), resulting in stronger complementarity and improved learning performance.

To validate this hypothesis, we performed both qualitative and quantitative analyses.

1. **Qualitative Visualization.** Intuitively, for a given input image, the less overlap there is between the regions attended by the two co-trained models, the more independent their respective "views" of the input become, and consequently, the more independent their predictions. Thus, for the same input image, we visualize the attention maps of the class token from the final Transformer block of both co-trained networks. As shown in Figure 3, CaPT's two co-trained networks exhibit markedly distinct attention patterns.

2. **Quantitative Measurement.** Given a labeled validation set $\{(x^{(i)}, y^{(i)})\}_{i=1}^{N}$, we collect the discrete predictions of the two co-trained models: $\hat{y}_1^{(i)} = h_1(x^{(i)})$ and $\hat{y}_2^{(i)} = h_2(x^{(i)})$. For each class $y$, we estimate the conditional mutual information as

$$\widehat{I}(H_1; H_2 \mid Y = y) = \mathrm{MI}\big(\{\hat{y}_1^{(i)}\}_{y^{(i)}=y}, \{\hat{y}_2^{(i)}\}_{y^{(i)}=y}\big), \tag{23}$$

and compute the overall conditional mutual information via

$$\widehat{I}(H_1; H_2 \mid Y) = \sum_y P(Y = y)\, \widehat{I}(H_1; H_2 \mid Y = y). \tag{24}$$

As shown in Figure 6, CaPT yields a substantially lower $\widehat{I}(H_1; H_2 \mid Y)$ compared to CLS, which is associated with a greater co-training accuracy improvement (see Table 7)[6].

These findings confirm that CaPT maintains stronger conditional independence of the two co-trained models' predictions given $Y$, and that this enhanced complementarity is a key factor in its co-training success.

## C  LLM Usage

We employed a large language model (LLM) to assist with the refinement of Section 1 and to verify the derivation process of Theorem 1.1.

## D  Loss Calculation for Labeled Data

Given a batch of labeled data containing $Q$ samples $\mathcal{X} = \{(x_j^l, y_j) : j \in (1, \ldots, Q)\}$. The supervised loss in CaPT is formulated—consistent with standard SSL practice—as the mean cross-entropy between the model's predicted class distribution and the ground-truth labels:

$$\mathcal{L}_s = \frac{1}{Q} \sum_{j=1}^{Q} H(y_j, p_m(y|x_j^l)), \tag{25}$$

where $p_m(y|x)$ is the predicted class distribution produced by the model for input $x$.

## E  CaPT with ResNet-50 Visual Encoder

We report in Table 8 the performance of CaPT on CIFAR-100 (Krizhevsky, 2009), STL10(Coates et al., 2011), and EuroSAT (Helber et al., 2019), using ResNet-50 (He et al., 2016) as the visual encoder for CLIP (Radford et al., 2021). Since CLIP's zero-shot classification performance is inferior when using ResNet-50 as the visual encoder compared to ViT-B/32, except on the EuroSAT dataset where their performance is similar, the overall performance of CaPT declines. Nevertheless, it still outperforms existing SSL methods.

## F  Experimental Details

We list the experimental configurations used in USB (Wang et al., 2022b) in Table 9. All pretrained ViT (Dosovitskiy, 2020) models are obtained from the links provided by USB. ViT-S-P2-32 indicates that we use ViT-Small with a patch size of 2 and an image size of 32. Since USB does not

---

[6]For fair comparison, our reproduced results of CLS are also based on FreeMatch (Wang et al., 2023) and USB benchmark. Additionally, we note that training CLS requires approximately 1.75× more time than CaPT.

Table 8: Accuracy (%) on CIFAR-100, STL10, and EuroSAT under USB. Different visual encoders for CLIP are used.

| Encoder | Dataset | CIFAR-100 | | STL10 | | EuroSAT | |
|---|---|---|---|---|---|---|---|
| | # Labels per Class | 2 | 4 | 4 | 10 | 2 | 4 |
| ViT-B/32 | CaPT | $84.83_{\pm 0.10}$ | $85.60_{\pm 0.07}$ | $96.07_{\pm 0.05}$ | $96.34_{\pm 0.05}$ | $96.60_{\pm 0.13}$ | $96.98_{\pm 0.11}$ |
| | Adapter-tuned CLIP | $74.90_{\pm 0.03}$ | $75.54_{\pm 0.02}$ | $96.86_{\pm 0.01}$ | $97.15_{\pm 0.01}$ | $93.83_{\pm 0.06}$ | $94.52_{\pm 0.4}$ |
| | CLIP | 65.10 | | 97.18 | | 49.46 | |
| RN50 | CaPT | $82.83_{\pm 0.35}$ | $84.85_{\pm 0.16}$ | $94.83_{\pm 0.07}$ | $95.17_{\pm 0.05}$ | $96.72_{\pm 0.15}$ | $96.87_{\pm 0.11}$ |
| | Adapter-tuned CLIP | $66.10_{\pm 0.08}$ | $69.44_{\pm 0.10}$ | $94.49_{\pm 0.02}$ | $94.58_{\pm 0.03}$ | $93.94_{\pm 0.03}$ | $94.48_{\pm 0.37}$ |
| | CLIP | 42.36 | | 94.37 | | 41.23 | |

Table 9: Training configurations and backbones in CaPT.

| Dataset | CIFAR-100 | STL10 | EuroSAT | SVHN | FGVCAircraft | ImageNet |
|---|---|---|---|---|---|---|
| Image Size | 32 | 96 | 32 | 32 | 224 | 224 |
| Model | ViT-S-P2-32 | ViT-B-P16-96 | ViT-S-P2-32 | ViT-S-P2-32 | ViT-B-P16-224 | ViT-B-P16-224 |
| Weight Decay | 5e-4 | | | | 5e-2 | |
| Labeled Batch size | 16 | | | | | 64 |
| Unlabeled Batch size | 16 | | | | | 64 |
| Learning Rate | 5e-4 | 1e-4 | 5e-5 | 5e-5 | 5e-4 | 3e-4 |
| Layer Decay Rate | 0.5 | 0.95 | 1.0 | 1.0 | 0.5 | 0.5 |
| Scheduler | $\eta = \eta_0 \cos(\frac{7\pi k}{16K})$ | | | | | |
| Model EMA Momentum | 0.0 | | | | | |
| Prediction EMA Momentum | 0.999 | | | | | |
| Weak Augmentation | Random Crop, Random Horizontal Flip | | | | | |
| Strong Augmentation | RandAugment (Cubuk et al., 2020) | | | | | |

include training configurations for FGVCAircraft (Maji et al., 2013), Flowers102 (Nilsback & Zisserman, 2008), StanfordCars (Krause et al., 2013), SUN397 (Xiao et al., 2010), DTD (Cimpoi et al., 2014), or SVHN (Netzer et al., 2011), we proceed as follows: for SVHN, we use the same ViT architecture as that used during EuroSAT (Helber et al., 2019) training; for the remaining datasets, we adopt the same ViT architecture as used during ImageNet (Deng et al., 2009) training, along with a unified training configuration across them. Consequently, Table 9 only reports CaPT's training configuration for the FGVCAircraft dataset.

Additionally, CLS requires two vision models; in our reimplementation, we utilize two versions of the pretrained ViT model: the vanilla ViT (Dosovitskiy, 2020) and the MAE-pretrained ViT (He et al., 2022).

## G  TRANSDUCTIVE ZERO-SHOT LEARNING

Table 10: Performances (%) of CaPT in T-ZSL setting.

| Dataset | CLIP | DebiasPL | **CaPT** |
|---|---|---|---|
| CIFAR-100 | 65.10 | 76.10 | **85.57** |
| EuroSAT | 49.46 | 69.98 | **70.17** |
| CIFAR-10 | 89.13 | 93.53 | **96.61** |
| ImageNet | 63.51 | 67.83 | **69.22** |

We emphasize another advantage of CaPT over previous SSL methods—its ability to leverage CLIP's zero-shot capability for transductive zero-shot learning (T-ZSL), bridging SSL and T-ZSL. Most prior SSL approaches cannot perform T-ZSL; to demonstrate CaPT's strength, we compare it with DebiasPL (Wang et al., 2022a), which selects high-confidence samples from CLIP's zero-shot predictions as the labeled dataset. Although CaPT can perform T-ZSL without modifications, to

obtain better performance, we select the top-$J$ highest-confidence samples per class (Huang et al., 2022) based on CLIP's predictions as the initial labeled data.

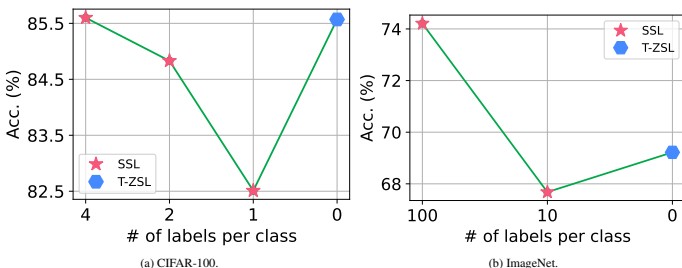

(a) CIFAR-100.

(b) ImageNet.

Figure 7: CaPT's performances in T-ZSL surpass its performances in SSL in some cases.

We set $J = S\lfloor\sqrt{C}\rfloor$, where $C$ is the number of classes in a dataset. Owing to CLIP's outstanding performance on CIFAR-10, we set $S = 300$ for CIFAR-10 and $S = 3$ for other datasets. As shown in Table 10, CaPT consistently outperforms others. Furthermore, Figure 7 visualizes CaPT's performance under SSL settings (with varying label counts) alongside its T-ZSL results. By leveraging unlabeled samples solely through CLIP's prior knowledge, CaPT even outperforms its performances in SSL in some cases, breaking the label dependency of SSL. This demonstrates that, in rare cases, the high-confidence dataset extracted from CLIP's priors can offer better supervision than the labeled training set used in SSL settings. Researchers can incorporate the high-confidence data selected in T-ZSL into the labeled training set for SSL to improve SSL performance.

Additionally, We present the prediction accuracy $A$ of the samples selected by CLIP for different values of $S$ in Figure 8. Because of CLIP's class bias, the number of selected samples $M$ might be less than $JC$. We use $M/JC$ to represent the uniformity of the selected samples. Finally, we use $A \times (M/JC)$ to represent the accuracy considering uniformity. The $S$ we choose is not necessarily optimal, and researchers can select better $S$ values based on the subfigures to achieve better T-ZSL performance.

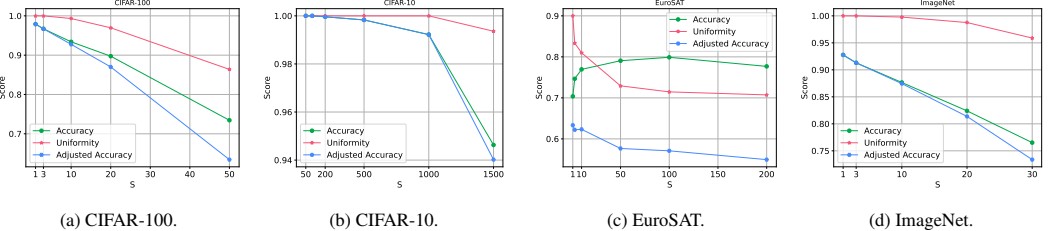

(a) CIFAR-100.  (b) CIFAR-10.  (c) EuroSAT.  (d) ImageNet.

Figure 8: The accuracy and uniformity of the selected labeled training set when $S$ takes different values, using ViT-B/32 as the visual encoder for CLIP.

## H CUSTOMIZED PROMPTS

When constructing class prompts, to obtain more representative prompts, we adopt the customized prompts proposed in CuPL (Pratt et al., 2023). Specifically, we first design large language model (LLM) prompts that guide the LLM to generate descriptions of the dataset categories (e.g., "What does a dog look like?"). Next, we feed these LLM prompts into the LLM to obtain prompts describing specific categories (e.g., "A dog looks like ..."). For each LLM prompt, we generate 10 different class prompts. The LLM prompts constructed for each dataset are listed in Table 11.

## I COMPARISON WITH OTHER SSL ALGORITHMS

In the main text, we reference several SSL algorithms, such as FlatMatch (Huang et al., 2023) and DebiasPL (Yao et al., 2022), but do not include them in the experimental section. This is due to

Table 11: LLM prompts used in CaPT.

| Dataset | LLM Prompts |
|---------|-------------|
| ImageNet | "Describe what a(n) {} looks like"
"How can you identify a(n) {}?"
"What does a(n) {} look like?"
"Describe an image from the internet of a(n) {}"
"A caption of an image of a(n) {}:" |
| CIFAR-10 | "What are the identifying characteristics of a(n) {}?"
"Describe a(n) {}:"
"Describe what a(n) {} looks like " |
| CIFAR-100 | "What are the identifying characteristics of a(n) {}?"
"Describe a(n) {}:"
"Describe what a(n) {} looks like "
"Describe a photo of a(n) {}" |
| DTD | "What does {} material look like?"
"What does a {} surface look like?"
"What does a {} texture look like?"
"What does a {} object look like?"
"What does a {} thing look like?"
"What does a {} pattern look like?" |
| EuroSAT | "Describe an aerial satellite view of {}"
"How does a satellite photo of a(n) {} look like"
"Visually describe a satellite view of a(n) {}" |
| STL10 | "What are the identifying characteristics of a(n) {}?"
"Describe a(n) {}:"
"Describe what a(n) {} looks like " |
| SVHN | "Describe a photo of the number {}"
"Describe a street sign of the number {}" |

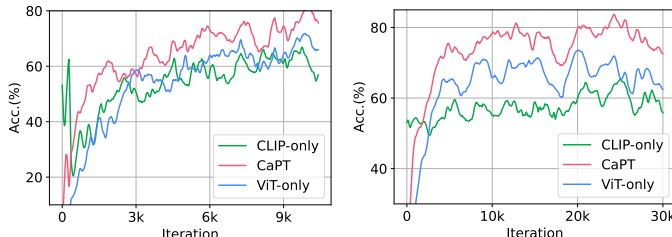

Figure 9: Pseudo label accuracy of three methods on CIFAR-100 with labeled data under $n_r$50 (left) and $i_r$30 (right).

the fact that these algorithms have not been evaluated on the USB benchmark. Their experiments commonly use Wide ResNet (Zagoruyko & Komodakis, 2016) trained from scratch as the training backbone. In this section, we adopt the same training setting and backbone as these algorithms for the unimodal network in our UPM, and compare CaPT with several SSL algorithms on the CIFAR-10 (Krizhevsky, 2009) dataset under different numbers of labeled samples. The results are shown in Table 12, where our method still achieves leading performance.

Table 12: Accuracy (%) on CIFAR-10 with varying numbers of labels per class. The best results are shown in **bold**, and the second-best in underline.

| Dataset | CIFAR-10 | | | |
|---|---|---|---|---|
| # Labels per Class | 1 | 4 | 25 | 400 |
| VAT (Miyato et al., 2018) | 20.19 | 25.34 | 58.97 | 89.49 |
| Mean Teacher (Tarvainen & Valpola, 2017) | 23.63 | 29.91 | 62.54 | 91.90 |
| MixMatch (Berthelot et al., 2019b) | 34.24 | 63.81 | 86.37 | 93.34 |
| ReMixMatch (Berthelot et al., 2019a) | 79.23 | 90.12 | 93.70 | 95.16 |
| UDA (Xie et al., 2020) | 65.47 | 89.38 | 94.84 | 95.71 |
| FixMatch (Sohn et al., 2020) | 75.21 | 92.53 | 95.14 | 95.79 |
| Dash (Xu et al., 2021) | 72.72 | 91.07 | 94.84 | 95.64 |
| MPL (Pham et al., 2021) | 76.45 | 93.38 | 94.24 | 95.45 |
| FlexMatch (Zhang et al., 2021a) | 86.15 | 95.03 | 95.02 | 95.81 |
| CLS (Yao et al., 2022) | - | 91.82 | 95.55 | 96.28 |
| SoftMatch (Chen et al., 2023) | - | 95.09 | 95.18 | 95.96 |
| DebiasPL (Wang et al., 2022a) | - | 94.60 | 95.40 | - |
| FreeMatch (Wang et al., 2023) | 91.93 | 95.10 | 95.12 | 95.90 |
| SequenceMatch (Nguyen, 2024) | - | 95.20 | 95.25 | 95.85 |
| FlatMatch (Huang et al., 2023) | 84.77 | 94.42 | 95.78 | 96.39 |
| RegMixMatch (Han et al., 2025) | 95.65 | 95.76 | 95.79 | 96.62 |
| **CaPT** | **95.93** | **96.35** | **96.57** | **96.80** |

## J CAPT'S RESULTS IN ADDITIONAL SCENARIOS WITH RESTRICTED SUPERVISION

Table 13: Accuracy (%) of CaPT on CIFAR-100 and EuroSAT under different levels of noisy labels ($n_r$) and class imbalance ($i_r$).

| Dataset | CIFAR-100 | | | | EuroSAT | | | |
|---|---|---|---|---|---|---|---|---|
| Condition | $n_r$25 | $n_r$50 | $i_r$15 | $i_r$30 | $n_r$25 | $n_r$50 | $i_r$15 | $i_r$30 |
| FreeMatch | 70.07 | 69.63 | 80.61 | 73.21 | 86.74 | 72.64 | 95.26 | 92.12 |
| RegMixMatch | 73.12 | 72.56 | 82.17 | 75.85 | 88.87 | 70.35 | 95.87 | 94.16 |
| **CaPT** | **78.88** | **76.41** | **85.10** | **84.40** | **95.60** | **88.54** | **96.96** | **96.80** |

Although advanced methods have been proposed to address low-label (Wang et al., 2023; Han et al., 2025), class-imbalanced (Lee et al., 2021), and noisy-label (Li et al., 2020) SSL regimes individually, few approaches tackle more than one of these challenges simultaneously. In realistic SSL scenarios (Gu et al., 2023), low-label conditions often imply that annotations are difficult to obtain—e.g., requiring expert knowledge—and that label reliability and class balance are harder to guarantee. Furthermore, label noise tends to affect classes unevenly, reducing the number of trustworthy annotations per class and amplifying class imbalance, which in turn degrades the performance of conventional SSL methods. By reducing SSL's dependence on labeled data, CaPT achieves more robust performance in practical, weakly supervised SSL settings.

Beyond low-label regime, we further evaluate CaPT in two additional label-constrained scenarios—SSL with noisy labels (NL) and class-imbalanced SSL (CI)—to demonstrate its effectiveness under limited supervision. The experimental setups for these tasks are defined as follows:

**NL Setup:** Each class is assigned 15 labeled samples, and $n_r\%$ denotes the fraction of labels that are randomly corrupted in each class.

**CI Setup:** Classes have varying numbers of labeled samples, gradually decreasing from 30 to fewer. We use $i_r$ to denote the imbalance ratio, defined as the ratio between the most and least labeled samples across classes.

Table 13 illustrates CaPT's consistently superior performance under constrained supervision. Figure 9 further presents the training curves of pseudo label accuracy for CaPT and its two variants—ViT-only (retaining only the ViT backbone) and CLIP-only (retaining only the CLIP component)—when supervised with noisy or class-imbalanced labeled data. Both variants yield suboptimal pseudo label accuracy: the ViT-only variant is hampered by constrained labeled data, while the CLIP-only variant suffers from few learnable parameters. CaPT combines their strengths by injecting reliable prior independent of labeled data into a robust ViT model, markedly enhancing unlabeled data use in SSL.

## K    FEW-SHOT METHODS

Despite methods like CLIP-Adapter (Gao et al., 2024), Tip-Adapter (Zhang et al., 2021b), and APE (Zhu et al., 2023) being able to adapt CLIP to few-shot classification efficiently by introducing only a minimal number of trainable parameters, this very parameter scarcity inherently confines their applicability to few-shot scenarios. As Table 14 illustrates, these advanced few-shot approaches still fall significantly short of CaPT's performance; accordingly, we do not extend our comparisons to additional few-shot methods.

Table 14: Comparison of CaPT and few-shot methods. Experiments are conducted with 2 labeled samples per class.

| Method | CIFAR-100 | EuroSAT |
|---|---|---|
| CLIP-Adapter | 68.43 | 80.22 |
| Tip-Adapter | 68.63 | 80.36 |
| APE-T | 68.88 | 80.47 |
| **CaPT** | **84.83** | **96.60** |

## L    PORTABILITY OF CAPT

Given the limited prior of CLIP and evolving nature of VLMs, Portability was a primary design consideration when developing CaPT as a training framework. We summarize its portability along three dimensions:

**Decoupled Co-Training Roles.**    The co-training scheme explicitly separates the provision of reliable prior (adapting VLMs) from the provision of strong learning capacity (training an SSL model). By disentangling these two roles, CaPT treats the prior and the learner as modular components that can be selected, improved, or replaced independently. In practice, this allows one to adopt stronger VLMs, employ more efficient VLM-adaptation strategies, or substitute alternative pseudo labeling

algorithms (e.g., replacing FixMatch (Sohn et al., 2020) with FreeMatch (Wang et al., 2023)) without modifying the rest of the pipeline (see Table 16 and Table 17). This modularity reduces reliance on any single component and enables CaPT to continuously benefit from future advances in both VLMs and SSL techniques.

**Efficient CLIP Tuning via Adapters.** Among the two major CLIP adapting strategies—prompt tuning (Zhou et al., 2022) and adapter tuning (Gao et al., 2024)—prompt tuning fails to reduce the gradient propagation flow (Wu et al., 2025), and thus does not effectively shorten training time. In contrast, our choice of adapter tuning significantly reduces training time and enhances the lightweight nature of CaPT.

**Architecture-Agnostic Adapter Design.** Many adapter-based methods (Yang et al., 2024; Khattak et al., 2023) insert adapters into internal layers and therefore require architecture-specific modifications to accommodate different VLM configurations (e.g., patch size or model scale). In contrast, our adapter-tuning strategy appends lightweight adapter modules to the model's output stage (i.e., at the tail of the architecture) rather than altering internal representations or layer layouts. By confining changes to the model's final stage, CaPT avoids structural modifications to the backbone and remains compatible with a wide range of VLMs. This simple, tail-attached design preserves the original model internals, reduces engineering overhead for different architectures, and retains the parameter-efficiency benefits of adapter tuning.

# M    POTENTIAL OVERLAP

Table 15: Zero-shot performance (%) of attribute-based CLIP (results directly taken from (Baron et al., 2024)).

| Dataset | With name | No name | Att.-finetuned (no name) |
|---|---|---|---|
| Dogs120 (Khosla et al., 2011) | 65.6 | 26.5 | 32.8 |
| OxfordPets (Parkhi et al., 2012) | 87.9 | 49.0 | 52.8 |
| CUB (Wah et al., 2011) | 62.6 | 19.8 | 24.0 |
| Flowers102 (Nilsback & Zisserman, 2008) | 72.3 | 24.4 | 39.4 |
| Food101 (Bossard et al., 2014) | 78.7 | 65.0 | 69.0 |

Although we conducted experiments on a wide range of fine-grained datasets (e.g., satellite imagery and texture recognition), one might still be concerned that the strong performance arises from potential overlap between CLIP's pre-training corpus and these datasets, leading to "memorization" of identical data or class names. We clarify as follows: CLIP is trained to align natural language descriptions with images in a shared semantic embedding space rather than to memorize class labels. Prior work (Pratt et al., 2023) has shown that replacing the fixed template prompt ``a photo of [classname]'' with more attribute-rich textual descriptions (e.g., ``A pink primrose generally has soft, pink petals with a yellow center'') can significantly improve zero-shot performance, indicating that CLIP captures attribute-based semantic alignment rather than merely matching class names. Furthermore, recent studies (Baron et al., 2024) demonstrate that decomposing class name concepts into attribute combinations allows CLIP to retain a non-negligible level of zero-shot capability **even when the class name itself is absent from the prompt**; fine-tuning CLIP with attribute-focused prompts can further enhance this zero-shot ability (see Table 15). Taken together, although we cannot completely rule out the possibility of a small amount of pre-training data overlap, the evidence from (1) robust performance gains with attribute-enriched prompts and (2) preserved zero-shot performance when only attribute descriptions are used provides a more compelling explanation that CLIP has learned attribute-level semantic alignment, enabling generalization across diverse tasks through attribute matching rather than simple memorization of labels or specific images in its pre-training corpus.

Table 16: Accuracy (%) of CaPT with various VLMs.

| Dataset | FGVCAircraft | | EuroSAT | |
|---|---|---|---|---|
| # Labels per Class | 5 | 10 | 2 | 4 |
| FreeMatch | 51.43 | 65.82 | 93.50 | 94.22 |
| RegMixMatch | 49.86 | _66.21_ | 95.75 | 96.39 |
| CaPT-CLIP | 50.12 | 64.33 | 96.60 | _96.98_ |
| CaPT-SigLIP | _53.33_ | 66.15 | _96.67_ | 96.90 |
| CaPT-CLIPA | **59.63** | **70.25** | **96.97** | **97.21** |
| CLIP | 18.97 | | 49.46 | |
| SigLIP | 15.32 | | 44.28 | |
| CLIPA | 40.06 | | 60.46 | |

Table 17: Accuracy (%) of CaPT with various pseudo label thresholding methods.

| Dataset | CIFAR100 | | EuroSAT | |
|---|---|---|---|---|
| # Labels per Class | 2 | 4 | 2 | 4 |
| FixMatch | 70.40 | 80.44 | 86.56 | 94.09 |
| CaPT-Fix | **77.87** | **83.66** | **93.53** | **96.45** |
| FlexMatch | 73.24 | 81.76 | 94.83 | 94.42 |
| CaPT-Flex | **78.55** | **84.26** | **97.01** | **96.88** |
| FreeMatch | 78.60 | 84.35 | 93.50 | 94.22 |
| CaPT-Free | **84.83** | **85.60** | **96.60** | **96.68** |

## N  FAILURE CASES OF CaPT

In Table 5, we show that CaPT outperforms previous methods on all evaluated fine-grained datasets except FGVCAircraft (Maji et al., 2013). We conjecture two reasons for this exception. First, CLIP exhibits relatively poor zero-shot performance on FGVCAircraft (18.97% zero-shot accuracy). Second, the version of CLIP used in our study carries a biased prior on this dataset that is more difficult to correct, because in our experiments we found that fine-tuning only increased CLIP's accuracy from 18.97% to around 32%. Since CaPT relies on correcting CLIP's prior to effectively leverage unlabeled samples, its performance is consequently constrained in this setting.

Nevertheless, we emphasize that the main contribution of CaPT lies in providing a framework for efficiently integrating CLIP into SSL, which can readily accommodate more advanced VLMs to achieve improved performance. Benefiting from CaPT's strong portability (Appendix L), we can efficiently incorporate advanced VLMs such as SigLIP (Zhai et al., 2023) and CLIPA (Li et al., 2023) into the CaPT pipeline. As shown in Table 16, when CaPT leverages priors from SigLIP and CLIPA[7], it achieves substantial performance gains over existing methods. Although SigLIP exhibits poorer zero-shot performance on FGVCAircraft (15.32% zero-shot accuracy), in our experiments we found that fine-tuning SigLIP can effectively correct its biased prior—increasing its accuracy from 15.32% to over 43%. This suggests that the more advanced SigLIP learns richer and more transferable representations, which can be more readily adapted to the downstream task through fine-tuning. As a result, CaPT can fully exploit the corrected prior, enabling effective utilization of unlabeled samples and strong overall performance on this dataset.

## O  CIFAR-10 (10) LABELED DATA VISUALIZATION

In Figure 1a, we present the performance of SSL algorithms on the CIFAR-10 dataset, where each class has one labeled sample, under labeled training sets with varying levels of prototypicality. Fix-

---

[7]We use ViT-B-16-SigLIP and ViT-L-14-CLIPA obtained from OpenCLIP (Cherti et al., 2023).

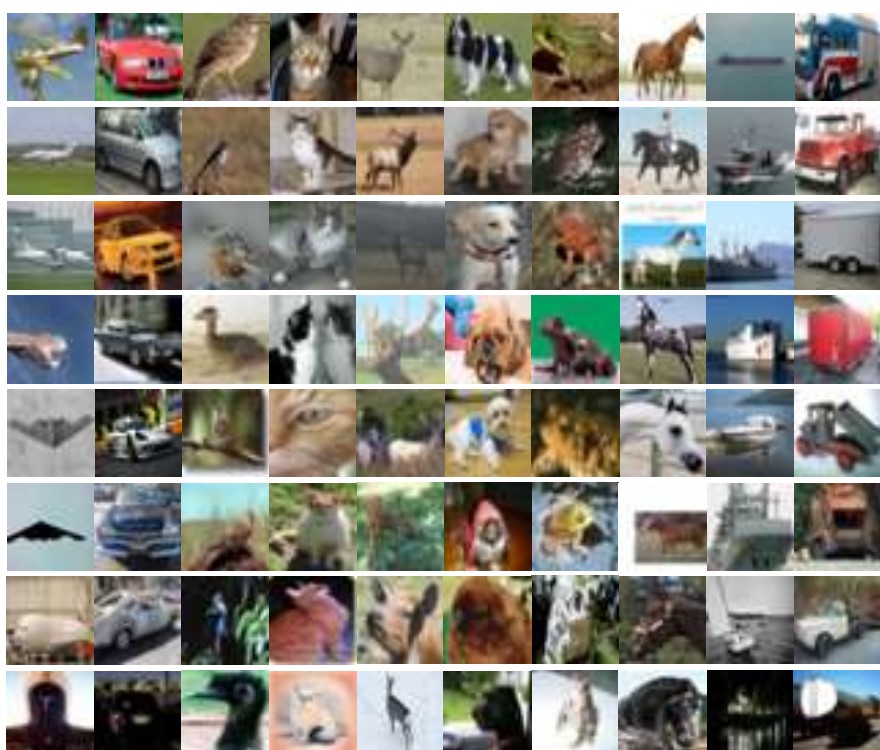

Figure 10: Labeled training data for the one-label-per-class experiment. Each row corresponds to a labeled training set, sorted from the most prototypical dataset (first row) to the least prototypical dataset (last row).

Match selects eight training sets with progressively decreasing class prototypicality through the ordering mechanism (Carlini et al., 2019). We visualize all the selected samples in Figure 10 as done in FixMatch, and in our experiments, we only use the first three labeled training sets.

## P    POTENTIAL IMPROVEMENTS

Although CaPT demonstrates excellent performance in practical weakly-supervised SSL settings. we point out several potential areas for improvement:

1. CaPT relies on refining the zero-shot capabilities of VLMs to better leverage unlabeled data in SSL. However, when a chosen VLM neither achieves satisfactory zero-shot performance on a target dataset nor admits an easily correctable prior, the VLM may become ineffective or even detrimental to SSL performance.

2. We propose an entropy-based weighting method to effectively allocate weights to the predictions of two classification models. Ideally, the weight assignment should reflect the models' accuracy on the validation set. However, our method failed to fully achieve this objective in some cases.

3. In T-ZSL, CaPT relies on CLIP's zero-shot capability to select high-confidence samples for each class as the labeled training set. However, when certain categories fail to have any samples selected due to CLIP's class bias, CaPT's performance in T-ZSL falls significantly behind SSL (e.g., EuroSAT).

4. CaPT cannot handle semi-supervised tasks outside of computer vision, such as time series and natural language processing.

