# OpenReview forum: "CLIP as a Prior Teacher: Breaking the Label Dependency in Semi-Supervised Learning"
_ICLR.cc/2026/Conference — Submitted to ICLR 2026_

### Official Review · Reviewer_Moze · 2025-10-25

**Soundness:** 3
**Presentation:** 2
**Contribution:** 3
**Rating:** 6
**Confidence:** 4

**Summary:**

The paper introduces a semi-supervised learning approach based on an asymmetric teacher–student scheme that uses CLIP as the guidance model. To mitigate known CLIP biases, the method combines consistency regularization with a lightweight fine-tuning strategy to keep compute overhead modest. A theoretical analysis explores learning with scarce labels, linking label quantity/quality to training dynamics and expected performance. Experiments cover multiple image-classification datasets, with ablations and visualizations that examine the contribution of each component.

**Strengths:**

- Coherent design and solid empirical gains: The integration of CLIP within an SSL pipeline is thoughtfully engineered; ablations suggest each component contributes meaningfully, and the reported results surpass the listed baselines.
- Clear exposition and positioning: The manuscript is generally easy to follow, and the related-work section situates the method well among comparable SSL approaches.
- Comprehensive experimentation: The empirical section is broad, includes analyses of the proposed regularization and fine-tuning, and emphasizes practical efficiency.
- Interesting theoretical motivation: The analysis connecting pseudo-label quality to labeled-data quality—and to how prototypical the labeled samples are—is insightful and adds value to the overall contribution.

**Weaknesses:**

- Questionable generality of the “framework” claim: CLIP differs meaningfully from modern VLMs, and CLIP itself is comparatively dated. Without evidence that the approach transfers cleanly to stronger/modern VLMs—or to other tasks—the contribution feels more like a targeted CLIP-based recipe than a broadly applicable framework. Demonstrating adaptability (e.g., with a second teacher family or a distinct task) would strengthen the novelty and impact.
- Scope limited to CLIP-based image classification: While effective in this setting, the study does not explore alternative tasks beyond classification. The paper does not claim multi-modality; however, discussing or lightly probing extensibility (even in a small-scale study) would strengthen the practical generality of the approach.
- Theory presentation could be clearer (minor but actionable):
    - Introduction: grammar around **line 50** needs a pass.
    - Symbols should be explicitly defined when first used: $\epsilon_n$ ,  $r$ , and  $\hat{y}$
    - Tightening these points would make the connection between assumptions and the training pipeline more transparent.

**Questions:**

- Teacher swapability: How readily can the teacher be replaced with stronger CLIP variants or contemporary vision–language models? Are there stability or calibration issues when doing so?
- Beyond classification: What modifications (if any) are needed to extend the method to detection/segmentation or image–text retrieval? Were any preliminary attempts made?
- Sensitivity to prompts and thresholds: How sensitive is performance to text-prompt choices, pseudo-label thresholds/temperatures (if applicable), and augmentation strength?

---

> ### Author Response · Authors · 2025-11-20
> **Response to Reviewer Moze**
>
> Thank you for acknowledging our *clear motivation, simplicity, reproducibility, thorough experiments, and the effectiveness* of the proposed method. We see that your main concerns relate to its generality, extensibility, and sensitivity. We address these points below.
>
> **Q1**: Can CLIP be replaced with stronger CLIP variants or contemporary vision–language models?
>
> **A1**: As shown in the table below, CaPT readily supports replacing CLIP with stronger CLIP variants; **framework generality was a primary design consideration** (see *Appendix L*). For more advanced VLMs (e.g., LLaVA), our co-training mechanism also permits integration into CaPT, although the original fine-tuning strategy may not be directly applicable. However, we **do not recommend** incorporating contemporary large VLMs into CaPT: these models typically contain many more parameters to retain language capabilities and far exceed the size of vision models commonly used in SSL. Consequently, even if we freeze the bulk of the VLM and only fine-tune lightweight adapters at the model tail to reduce backward gradient propagation flow, **the forward computational cost of the VLM can still be substantially higher than the ViT forward+backward cost, raising practical and cost challenges**.
>
> |  | CIFAR100 |  |  | EuroSAT |  |  | FGVCAircraft |   |
> |---|---|---|---|---|---|---|---|---|
> |  | 2 | 4 |  | 2 | 4 |  | 5 | 10  |
> | FreeMatch | 78.60 | 84.35 |  | 93.50 | 94.22 |  | 51.43 | 65.82  |
> | CaPT-CLIP | 84.83 | 85.60 |  | 96.60 | 96.98 |  | 50.12 | 64.33  |
> | CaPT-SigLIP | 87.42 | 87.64 |  | 96.67 | 96.90 |  | 53.33 | 66.15  |
> | CaPT-CLIPA | **89.98** | **90.23** |  | **96.97** | **97.21** |  | **59.63** | **70.25**  |
>
>
> **Q2**: Beyond classification.
>
> **A2**: The CaPT framework is primarily designed for image classification and cannot be directly applied to detection, segmentation, or image–text retrieval tasks. However, its core principles are general: to extend CaPT to detection or segmentation, one could replace the original modules A and B with corresponding detection/segmentation models and models with zero-shot capabilities (e.g., F-VLM, ZegCLIP). Since this work focuses on image classification, we have not conducted preliminary experiments in these other domains. Given that annotation is typically even more scarce in detection and segmentation tasks, we believe this represents a promising direction for future research.
>
> **Q3**: Sensitivity to prompts and thresholds.
>
> **A3**:
>
> **Prompts**. As shown in the table below, since we fine-tune CLIP and have access to a large number of trainable samples, using more advanced text prompting methods (e.g., CuPL) **has little impact on our results**—it mainly affects CLIP’s zero-shot or few-shot accuracy.
>
> |  | CIFAR100 | EuroSAT  |
> |---|---|---|
> | CaPT w/o CuPL | 84.81 | 96.55  |
> | CaPT w CuPL | 84.83 | 96.60  |
>
>
> **Thresholds**. Thanks to the **generality** of our framework, we can also replace the pseudo label thresholding with that of different SSL methods, as shown in the table below. CaPT **consistently improves performance** across various pseudo label threshold strategies, and naturally, more advanced pseudo labeling techniques can achieve even better results.
>
> |  | CIFAR100 |  |  | EuroSAT |   |
> |---|---|---|---|---|---|
> | Labels per Class | 2 | 4 |  | 2 | 4  |
> | FixMatch | 70.40 | 80.44 |  | 85.56 | 94.09  |
> | CaPT-Fix | **77.87** | **83.66** |  | **93.53** | **96.45**  |
> | FlexMatch | 73.24 | 81.76 |  | 94.83 | 94.42  |
> | CaPT-Flex | **78.55** | **84.26** |  | **97.01** | **96.88**  |
> | FreeMatch | 78.60 | 84.35 |  | 93.50 | 94.22  |
> | CaPT-Free | **84.83** | **85.60** |  | **96.60** | **96.88**  |
>
> **Augmentation strength**. To ensure **fairness**, our data augmentation strategy follows standard practice in mainstream SSL methods: weak augmentation uses RandomCrop and RandomHorizontalFlip, while strong augmentation adopts RandAugment, with augmentation strengths **kept identical to prior SSL work**. Feature-level augmentation is applied via Mixup, whose strength is controlled by the mixing intensity α. The effect of different α on CaPT is shown in the table below.
>
> | α | 0 | 0.1 | 0.2 | 0.5 | 1  |
> |---|---|---|---|---|---|
> | CIFAR100 | 84.26 | 84.64 | **84.83** | 84.76 | 84.22  |

---

> ### Author Response · Authors · 2025-11-24
> **Looking forward to your response**
>
> Dear Reviewer Moze,
>
> Thanks for your careful review and constructive comments. We have revised the paper and provided a point-by-point response addressing your concerns. Please let us know if you still have concerns regarding the paper.

---

### Official Review · Reviewer_rE4S · 2025-10-29

**Soundness:** 2
**Presentation:** 2
**Contribution:** 2
**Rating:** 2
**Confidence:** 4

**Summary:**

This paper attempts to address a well-known problem in SSL: the model's performance is heavily dependent on the quantity and quality of the limited labeled data. The authors claim to break this dependency by introducing CaPT. The core idea is to concurrently train a fft unimodal network on images and a parameter-efficiently fine-tuned (PEFT) CLIP model. These two models supervise each other via an entropy-weighted co-pseudo label. The results show that CaPT achieved state-of-the-art (SOTA) performance across multiple SSL benchmarks, especially in extremely low-label settings.

**Strengths:**

1. The authors have tested CaPT on a wide range of benchmarks, including USB, ImageNet, and several fine-grained datasets, covering various scenarios of label scarcity, label noise, and class imbalance.

**Weaknesses:**

1. The core contribution of this paper is severely overclaimed. The CaPT, is, in my opinion, nothing more than a simple combination of several existing ideas, such as co-training, adapter-tuning, and mixup.
2. The authors make the assertion that their work breaks the label dependency. In reality, they have merely replaced the dependency on high-quality labels with a dependency on high-quality CLIP prior. This is laid bare in Appendix N: when CLIP performs poorly on the FGVCAircraft dataset, CaPT's performance is low as well.
3. Entropy-based weighting is naive. Did you explore any other, more robust weighting strategies?

**Questions:**

See in Weaknesses.

---

> ### Author Response · Authors · 2025-11-20
> **Response to Reviewer rE4S--Part 1**
>
> Thank you for acknowledging the *comprehensiveness* of our experiments. We hope the following responses address your concerns regarding our work.
>
> **Q1**: The label dependence formalized in this paper is a well-known problem in SSL.
>
> **A1**: We may not have clearly articulated our underlying motivation, and the revised version now provides a clearer explanation. As acknowledged by *Reviewer z82e, Moze and Uzf4*, we clarify that the label dependence formalized in Theorem 1.1 is **not** the well-known notion that SSL performance depends on labeled data, but rather the **unique dependence** of **unlabeled data on labeled data**. This is a critical issue **overlooked** in prior SSL research: although SSL ostensibly depends on two data sources, the utility of unlabeled samples is tightly **coupled** to labeled data. **Paradoxically and unexpectedly, as the supervision from labeled data deteriorates, SSL instead becomes more dependent on that limited labeled data and can even fail to benefit from unlabeled data when the labeled set is sufficiently poor**. This explains the abrupt performance drop observed in *Figure 1a*.
>
> Breaking this **unique dependence** thus constitutes a genuine central challenge for SSL under **practical constraints** like label scarcity, label noise, and class imbalance—**one that prior work has not explicitly addressed**.
>
>
> **Q2**: CaPT’s core contribution is merely a simple combination of existing ideas.
>
> **A2**: We kindly request the reviewer to reconsider CaPT from the **perspective of the overall framework and research goals**—rather than focusing solely on **individual technical components**. As *Reviewer z82e* also recognizes, our aim is not to propose another fine-tuning or data augmentation trick, but to **answer when and why CLIP priors should be incorporated into SSL and to design a broadly applicable, robust, and efficient “VLM + SSL” training framework**.
>
> Our contributions are twofold. Theoretically, we provide the first formalization and analysis of SSL’s label dependence, **offering a principled justification for introducing external priors**. On the systems side, CaPT’s design is **principled**, **not a casual assembly of techniques**: deliberately chosen co-training naturally corrects CLIP’s biased prior and satisfies model learning capacity requirements while preserving compatibility across SSL methods and VLMs. Other components are **integrated with clear, complementary purposes**—to improve generality (adapter tuning), efficiency (lightweight adapters, Mixup), and reliability (adapter tuning and entropy-based weighting).
>
> Thus, CaPT’s core contribution lies in **validating a scalable, future-proof training framework for integrating VLM priors into SSL—not in stitching together known tricks into a ad-hoc solution**.

---

> ### Author Response · Authors · 2025-11-20
> **Response to Reviewer rE4S--Part 2**
>
> **Q3**: CaPT merely replaces the dependency on labels with a dependency on high-quality CLIP prior.
>
> **A3**: We would like to clarify that CaPT is **by no means** merely replacing low-quality labels with high-quality CLIP priors—this **misinterprets** our framework’s core mechanism.
> - Take the EuroSAT as an example (see the table below): under the same ViT backbone, the SSL model trained solely with CLIP’s priors achieves an accuracy of **70.17**, which is **far lower** than the model trained with only one labeled sample per class (**90.12**). This directly proves that the **supervision provided by CLIP’s raw prior is far inferior to that provided by even extremely scarce labeled data**. Importantly, even in this setting, introducing CLIP via CaPT does not harm performance; instead, CaPT **yields significant improvements** (*Table 3*).
>
> | Supervision | CLIP | 1 label per class | 2 labels per class | 4 labels per class  |
> |---|---|---|---|---|
> | Accuracy | 70.17 | 90.12 | 93.50 | 94.22  |
>
> - Therefore, not one-way injection, the key lies in **bidirectional mutual learning** (see *Figure 2d*). Mutual learning lets the **fully fine-tuned ViT supervise and progressively adapt CLIP’s general representations to the downstream distribution**. Thus, whether CLIP can boost SSL is determined not by its static zero-shot accuracy, but by the **dynamic interaction between the two models during co-training**. *Appendix N* provides clear evidence: although SigLIP has lower zero-shot accuracy (**15.32**), its **more adaptable general knowledge** enables CaPT–SigLIP to **improve** SSL performance on FGVC (**51.43 → 53.33**), whereas CaPT–CLIP degrades performance (**51.43 → 50.12**). This also explains why CLIP helps on EuroSAT: as *Figure 5* shows, the dynamic interaction adapts CLIP’s initially biased prior toward a more balanced prior, providing more reliable guidance for SSL.
> - *Appendix L* further demonstrates CaPT’s **generality**: the framework allows CLIP to be **readily replaced with more advanced, task-adaptive VLMs (far outperforming the CLIP model adopted in this work) [1], eliminating any "dependency on a single CLIP prior" and enabling performance to improve as VLMs evolve**.
>
> **References**
>
> [1] Ilharco, G., Yu, T., Zhuang, L., Jain, A., & Song, H. F. (2021). *OpenCLIP: An Open Reimplementation of CLIP*. GitHub repository: https://github.com/mlfoundations/open_clip
>
> **Q4**: Entropy-based weighting is naive, any other more robust weighting strategies?
>
> **A4**: In prior co-training methods, supervision is typically exchanged by **simply swapping pseudo labels**. In our **asymmetric modalilty** co-training, however, **structural asymmetries** between modalities and models result in **pseudo labels of uneven and dynamically changing quality**. Blindly adopting the traditional exchange strategy therefore yields suboptimal performance (see table below).
>
> Beyond the entropy-based weighting described in the paper, we have explored using **stability** to capture dynamic pseudo label reliability. Specifically, for each co-training model, we maintain an EMA of gradient directions and use the cosine similarity between the current update and the historical gradient EMA as a **stability-based weighting metric**. However, this strategy did not yield satisfactory performance. Although entropy-based weighting is relatively simple, it can achieve better performance. **We are currently continuing to explore more adaptive weighting strategies tailored to asymmetric modalities**.
>
> |  | CIFAR100 | EuroSAT  |
> |---|---|---|
> | Simply swapping | 74.60 | 95.55  |
> | Stability weighting | 82.66 | 95.46  |
> | Entropy weighting | **84.83** | **96.60**  |

---

> ### Author Response · Authors · 2025-11-24
> **Looking forward to your feedback**
>
> Dear Reviewer rE4S,
>
> Thanks for your valuable suggestions about our paper. We have revised the paper and provided a point-by-point response addressing your concerns. If our replies have resolved your concerns, we would be grateful if you would consider raising score. Should any questions remain, we warmly invite further discussion and will gladly supply any additional information you need.

---

### Official Review · Reviewer_z82e · 2025-10-31

**Soundness:** 3
**Presentation:** 3
**Contribution:** 3
**Rating:** 6
**Confidence:** 4

**Summary:**

This paper introduces CaPT (CLIP as a Prior Teacher), a novel semi-supervised learning framework that leverages the strong generalization ability of CLIP to reduce the dependency of SSL methods on labeled data.
The key idea is to treat CLIP as a prior teacher, combining its zero-shot semantic knowledge with a unimodal visual learner through an asymmetric co-training mechanism.
The paper also provides theoretical insights into label dependency in SSL and demonstrates significant performance gains on standard benchmarks under extreme low-label conditions.

**Strengths:**

1. The paper formalizes label dependency in SSL and clearly articulates why existing methods fail when labeled data are extremely scarce.

2. The asymmetric co-training between CLIP and the visual model is simple yet well-motivated, enabling complementary learning between prior knowledge and data-driven adaptation.

3. Extensive experiments on CIFAR, STL, and ImageNet subsets show consistent improvements over strong SSL baselines (FixMatch, FreeMatch, RegMixMatch, etc.), especially in 1-shot and 2-shot settings.

**Weaknesses:**

1.	While some ablations are included, it would be useful to see results with other multimodal priors (e.g., SigLIP, EVA-CLIP) to confirm generality.
	2.	The paper focuses on SSL baselines but could better position itself against few-shot or distillation-based methods.

**Questions:**

See Weaknesses.

**Details Of Ethics Concerns:**

See Weaknesses.

---

> ### Author Response · Authors · 2025-11-20
> **Response to Reviewer z82e**
>
> Thank you for acknowledging our *clear motivation, simplicity, thorough experiments, and the effectiveness* of the proposed method. We understand that your main concerns are generality and comparison with broader methods. We now address your questions.
>
> **Q1**: CaPT's generality.
>
> **A1**: We have **prioritized generality as a core design principle** of our training framework (see *Appendix L*). This deliberate focus enables strong adaptability: our framework allows for the **seamless replacement of CLIP with more advanced VLMs**, which consistently yields improved performance as shown in *Appendix N* of the paper and the table below. This directly highlights CaPT’s primary contribution: **it stands as a general-purpose, end-to-end framework that effectively integrates VLMs into SSL while naturally accommodating future advances in VLMs**.
>
> |  | CIFAR100 |  |  | EuroSAT |  |  | FGVCAircraft |   |
> |---|---|---|---|---|---|---|---|---|
> |  | 2 | 4 |  | 2 | 4 |  | 5 | 10  |
> | FreeMatch | 78.60 | 84.35 |  | 93.50 | 94.22 |  | 51.43 | 65.82  |
> | CaPT-CLIP | 84.83 | 85.60 |  | 96.60 | 96.98 |  | 50.12 | 64.33  |
> | CaPT-SigLIP | 87.42 | 87.64 |  | 96.67 | 96.90 |  | 53.33 | 66.15  |
> | CaPT-CLIPA | **89.98** | **90.23** |  | **96.97** | **97.21** |  | **59.63** | **70.25**  |
>
>
> **Q2**: Position CaPT against few-shot or distillation-based methods.
>
> **A2**:
>
> **Few-shot**. As shown in *Appendix K* and the table below, while few-shot methods can efficiently adapt CLIP to few-shot classification by introducing only a minimal number of trainable parameters, this very parameter scarcity inherently limits their performance in SSL settings **with more samples available for learning**.
>
> |  | CIFAR100 | EuroSAT  |
> |---|---|---|
> | CLIP-Adapter | 68.43 | 80.22  |
> | Tip-Adapter | 68.63 | 80.36  |
> | APE-T | 68.88 | 80.47  |
> | CaPT | **84.83** | **96.60**  |
>
> **Distillation**. Thanks to your insightful reminder, we have taken note of the contemporaneous work *DHO* [1], which employs a distillation-based approach to apply CLIP to SSL. We conducted a preliminary analysis between co-training and distillation from both conceptual and empirical perspectives.
>
> Conceptually, distillation relies on a unidirectional flow of information, where the **student fully trusts the teacher’s predictions**. In contrast, co-training is built upon the assumption that **the teacher’s predictions are not perfect and should be refined** through bidirectional mutual learning. This assumption aligns well with the fact that CLIP often carries **biased priors** in downstream domains (see *Figure 5*), making co-training naturally suited for correcting these biases.
> Empirically, we implemented an ablation of CaPT in which we retain only the CLIP→vision model unidirectional flow (i.e., approximating a distillation setup). As shown in the table below, this modification leads to a noticeable performance drop.
> Based on the above analysis, we *tentatively* conclude that the co-training mechanism is more suitable for SSL scenarios.
>
> |  | CIFAR100 | EuroSAT  |
> |---|---|---|
> | CaPT-Dist | 83.95 | 95.11  |
> | CaPT | **84.83** | **96.60**  |
>
> **References**
>
> [1] Kang et al., 2025. Simple yet Effective Semi-supervised Knowledge Distillation from Vision-Language Models via Dual-Head Optimization.

---

> ### Author Response · Authors · 2025-11-24
> **Looking forward to your response**
>
> Dear Reviewer z82e,
>
> Thanks for your careful review and constructive comments. We have revised the paper and provided a point-by-point response addressing your concerns. Please let us know if you still have concerns regarding the paper.

---

### Official Review · Reviewer_Uzf4 · 2025-11-01

**Soundness:** 2
**Presentation:** 2
**Contribution:** 2
**Rating:** 4
**Confidence:** 4

**Summary:**

The paper proposes a semi-supervised learning pipeline that leverages CLIP’s prior knowledge in a co-training framework where CLIP is updated with parameter-efficient fine-tuning (e.g., adapters) rather than full model tuning. Experiments on common SSL benchmarks show performance gains.

**Strengths:**

1. The paper is build on clear motivation with supporting theory showing pseudo-label error grows with prototype bias and fewer labels, formalizing a well-known intuition.

2. The paper proposes a practical strategy to incorporate CLIP in SSL that balances efficiency (adapter tuning, feature-level Mixup) and reliability (co-training + entropy-weighted labels)

**Weaknesses:**

1.Domain dependence of CLIP priors: where CLIP is strong (e.g., natural images like CIFAR), gains are intuitive; where CLIP is weaker or off-distribution (e.g., EuroSAT and many medical domains), benefits may diminish and are harder to guarantee.


2. Technical contributions feel like a careful combination of known pieces (co-training, PEFT adapters, entropy-weighted pseudo-labels, Mixup)

**Questions:**

NA

---

> ### Author Response · Authors · 2025-11-20
> **Response to Reviewer Uzf4**
>
> Thanks for acknowledging our *clear motivation, practicality and effectiveness* of the proposed method. We see that your main concerns are CLIP’s domain dependence and our method design. We now address your questions.
>
> **Q1**: CaPT’s performance on off-distribution domains (e.g., EuroSAT and medical datasets) are harder to guarantee due to CLIP’s domain dependence.
>
> **A1**:
> - As the **first** work to formally leverage VLM priors to mitigate label dependence in SSL, we deliberately chose the **original CLIP** as the prior to highlight the improvements that VLMs can bring to SSL. While CLIP’s priors are limited, **VLMs are rapidly evolving**, and many **have already attained strong zero-shot performance (far outperforming the CLIP model adopted in this work) in OOD domains [1]**. Critically, our framework is designed with **generality as a core principle** (*Appendix L*), it can **readily** incorporate more advanced VLMs to **remain up to date**. This underscores our principal contribution: **designing a general framework rather than proposing a fixed SSL algorithm**.
> - As a method that leverages CLIP priors to enhance SSL, our approach is inevitably constrained by CLIP’s domain knowledge. However, our **targeted design** has mitigated this limitation to the greatest extent possible: we **keep CLIP fine-tunable and allow it to receive supervision from the fully fine-tuned ViT via mutual learning**, thereby markedly reducing reliance on its original domain priors. For example, *Figure 5* shows that CLIP’s raw prior performs very poorly on EuroSAT. Nevertheless, mutual learning and fine-tuning enable **CLIP’s pretrained general representations to successfully adapt to the downstream task**, allowing CaPT to still benefit from CLIP’s **adapted general knowledge** (we clarify that CaPT achieves **excellent performance on EuroSAT**, see Footnote 3 in paper). The FGVC example in *Appendix N* provides additional validation.
>
> **References**
>
> [1] Ilharco, G., Yu, T., Zhuang, L., Jain, A., & Song, H. F. (2021). *OpenCLIP: An Open Reimplementation of CLIP*. GitHub repository: https://github.com/mlfoundations/open_clip
>
> **Q2**: Technical contributions feel like a careful combination of known pieces.
>
> **A2**: We clarify that our work is **not about pursuing SOTA through the mere assembly of advanced techniques**. As the first study to formally elucidate *when and why* CLIP priors should be incorporated into SSL, our core focus lies in **developing a general "VLM + SSL" training framework**. The advanced techniques we adopt are used as **pragmatic tools to improve the framework’s efficiency, reliability, and generality—not as an end in themselves**. This directly highlights CaPT’s primary contribution: **establishing a scalable, future-proof framework for integrating VLMs into SSL**.

---

> ### Author Response · Authors · 2025-11-24
> **Looking forward to your feedback**
>
> Dear Reviewer Uzf4,
>
> Thanks for your careful review and constructive comments. We have revised the paper and provided a point-by-point response addressing your concerns. If our replies have resolved your concerns, we would be grateful if you would consider raising your score. Should any questions remain, we warmly invite further discussion and will gladly supply any additional information you need.

---

### Official Review · Reviewer_u1wY · 2025-11-04

**Soundness:** 2
**Presentation:** 2
**Contribution:** 2
**Rating:** 2
**Confidence:** 3

**Summary:**

This paper focuses on CLIP-based semi-supervised learning. First, the paper demonstrates through theoretical and empirical analysis that performance is limited by the quantity and quality of labeled data. Then, it proposes a new method called CLIP as a Prior Teacher (CaPT), encompassing three modules: a pseudo-label module based on ViT, an adapter tuning module, and an ensemble module that combines the predictions of the first two modules. Experimental results validate the effectiveness of the proposed method.

**Strengths:**

The studied problem of semi-supervised learning with CLIP is an important and interesting research topic.

The experimental results are very comprehensive and validate the effectiveness of the proposed method.

**Weaknesses:**

The proposed approach seems to be a direct combination of the FixMatch approach (module A) and the parameter-efficient fine-tuning approach (module B). Although the co-training technique is interesting, directly combining off-the-shelf approaches may weaken the paper's contributions.

The paper's layout can be improved. First, it is unusual to include a theorem in the introduction. Additionally, it is not rigorous to directly call the different modules "A," "B," and "C." Additionally, there are typos, such as "though" instead of "through" in line 229. Furthermore, the augmentation in Eq. (2) is an addition. However, this is not always true, as many augmentations cannot be implemented by simply adding a feature to another vector or tensor.

Although the co-training scheme is effective, involving a ViT and a CLIP model together is much more complex than the compared methods.

Theorems 1 seem irrelevant to the motivation and the proposed approach. First, it is obvious that the classifier's performance will be inferior with less data, without the need for any theoretical analysis. Second, accuracy is the nearest-prototype pseudo-label error, which is different from the classification model. Third, a larger upper bound does not necessarily indicate a smaller label error.

**Questions:**

Please see "weaknesses".

---

> ### Author Response · Authors · 2025-11-20
> **Response to Reviewer u1wY--Part 1**
>
> Thank you for acknowledging our *novelty, thorough experiments, method effectiveness*, and your **helpful suggestions on organization**. We understand that your main concerns lie in our motivation and method design. We address them below.
>
> **Q1**: Directly combining FreeMatch and PEFT may weaken the paper's contributions.
>
> **A1**: We have **intentionally** kept the combination of SSL methods and VLM fine-tuning strategies simple.
> - As the **first** work to formally elucidate *when and why* CLIP should be incorporated into SSL, we deliberately avoid auxiliary engineering tricks. This minimal design lets us isolate the causal effect of VLM priors on the utility of unlabeled data, **making the value of CLIP in mitigating label dependence in SSL far more explicit.**
> - As shown in *Appendix L*, this simplicity also ensures **portability**. CaPT is designed as a **framework** rather than a fixed algorithm: its modular components allow us to swap FixMatch for other SSL methods (e.g., FreeMatch) and to replace CLIP with stronger VLMs (see table below). **Such extensibility is particularly crucial given the rapid evolution of VLMs.**
> - Finally, as acknowledged by *Reviewer z82e*, a **simple yet well-motivated** design that produces **strong results** is itself valuable because it improves reproducibility and adoptability.
>
> |  | CIFAR100 |  |  | EuroSAT |  |  | FGVCAircraft |   |
> |---|---|---|---|---|---|---|---|---|
> |  | 2 | 4 |  | 2 | 4 |  | 5 | 10  |
> | FreeMatch | 78.60 | 84.35 |  | 93.50 | 94.22 |  | 51.43 | 65.82  |
> | CaPT-CLIP | 84.83 | 85.60 |  | 96.60 | 96.98 |  | 50.12 | 64.33  |
> | CaPT-SigLIP | 87.42 | 87.64 |  | 96.67 | 96.90 |  | 53.33 | 66.15  |
> | CaPT-CLIPA | **89.98** | **90.23** |  | **96.97** | **97.21** |  | **59.63** | **70.25**  |
>
>
> |  | CIFAR100 |  |  | EuroSAT |   |
> |---|---|---|---|---|---|
> | Labels per Class | 2 | 4 |  | 2 | 4  |
> | FixMatch | 70.40 | 80.44 |  | 85.56 | 94.09  |
> | CaPT-Fix | **77.87** | **83.66** |  | **93.53** | **96.45**  |
> | FlexMatch | 73.24 | 81.76 |  | 94.83 | 94.42  |
> | CaPT-Flex | **78.55** | **84.26** |  | **97.01** | **96.88**  |
> | FreeMatch | 78.60 | 84.35 |  | 93.50 | 94.22  |
> | CaPT-Free | **84.83** | **85.60** |  | **96.60** | **96.88**  |
>
> **Q2**: Layout suggestions, including placement of the theorem, module naming, typos, and augmentation notation
>
> **A2**:
>
> **Placement of the theorem.** Introducing a theoretical result in the Introduction is indeed unconventional; however, this is precisely what sets our work apart from other theorem-integrated works. Our theoretical finding are placed early because it directly **underpin our motivation**—whereas prior works primarily introduce theorem to **justify their proposed methods**.
>
> **Module naming.** We have replaced the previous names with more appropriate ones in the revised version.
>
> **typo.** “through” is **not** a typo. Our intended meaning is that obtaining strong and weak predictions **by** fully fine-tuning CLIP is time-consuming; this observation motivated our design of PEFT and feature-augmented consistency regularization.
>
> **Augmentation notation.** We have updated the augmentation notation in the revised version.
>
> **Q3**: Although co-training is effective, co-training a ViT and a CLIP model is relatively more complex.
>
> **A3**: We agree that co-training a ViT and a CLIP model is indeed more complex than training a single model. However, this complexity is justified by following key considerations:
> - As shown in *Table 6*, both components are **indispensable**: fine-tuning CLIP to correct its prior and fully training a model with strong learning capacity are each necessary for the observed gains.
> - Our *deliberate design* ensures that despite training two models, the overall resource overhead **does not incur a significant increase** compared with training a single SSL model alone (*Table 4*).
> - This co-training framework brings **substantial improvements** to SSL, which further validates that **the marginal complexity is well worth the performance return**.

---

> ### Author Response · Authors · 2025-11-20
> **Response to Reviewer u1wY--Part 2**
>
> **Q4**: Theorem 1.1 appears unrelated to the paper; some suggestions on rigor.
>
> **A4**: We may not have clearly articulated our underlying motivation, and the revised version now provides a clearer explanation. We would like to clarify the connection between Theorem 1.1 and our proposed method. As acknowledged by *Reviewer z82e, Moze and Uzf4*, the label dependence formalized in Theorem 1.1 does **not** refer to the **reliance of SSL performance on labeled data**, but specifically to the **unique dependence** of **unlabeled data on labeled data**. Concretely, Theorem 1.1 shows that the poorer the supervision from labeled data, the higher the upper bound of pseudo label error for unlabeled samples. This indicates that, although SSL ostensibly relies on two data sources, the utility of unlabeled samples is **tightly coupled** to the labeled set. **Paradoxically and unexpectedly, as labeled supervision deteriorates, SSL can become even more dependent on that limited labeled data and may ultimately fail to benefit from unlabeled data**—this explains the abrupt performance drop observed in *Figure 1a*.
>
> Our key insight is therefore to introduce **alternative unlabeled data utilization mechanisms** that break this **unique dependence** of unlabeled data on labeled data. We hope the above clarifies the first two concerns. Finally, we appreciate your rigorous observation regarding the error upper bound, and we have revised it to a more rigorous expression in the revised version.

---

> ### Author Response · Authors · 2025-11-24
> **Looking forward to your feedback**
>
> Dear Reviewer u1wY,
>
> Thanks for your careful review and constructive comments. We have revised the paper and provided a point-by-point response addressing your concerns. If our replies have resolved your concerns, we would be grateful if you would consider raising your score. Should any questions remain, we warmly invite further discussion and will gladly supply any additional information you need.

---

> > ### Comment · Reviewer_u1wY · 2025-11-28
> > **Thanks for the rebuttal!**
> >
> > Thanks for the rebuttal, which addresses parts of my concerns well. Therefore, I will raise my score.

---

### Meta-Review · Area_Chair_cAn1 · 2025-12-31

**Summary:**

This paper proposes CaPT, a framework for Semi-Supervised Learning (SSL) that integrates Vision-Language Models (VLMs) via asymmetric co-training. The authors aim to solve "label dependency" by using a PEFT-tuned CLIP model as a "Prior Teacher" to assist a standard unimodal ViT.

The paper demonstrates exceptional empirical performance, particularly in extreme low-label settings (e.g., +21.38% on CIFAR-100). However, the central reason for the Reject recommendation is the limited technical contribution. The reviewers (particularly R4 and R2) identified that the framework is a combination of existing components—FixMatch, CLIP, PEFT (adapters), and co-training—without introducing a fundamentally new learning paradigm or architectural innovation. For a top-tier venue like ICLR, the consensus is that strong empirical results on existing benchmarks do not sufficiently compensate for a lack of algorithmic novelty.

**Reviewer Concerns:**

### Addressed by Rebuttal:
*   The authors provided extensive additional experiments using newer VLMs (SigLIP, CLIPA), proving the framework is portable and scales with better priors.
*   The rebuttal clarified that bidirectional learning allows the model to adapt even when CLIP's zero-shot performance is low (e.g., EuroSAT).
*  The authors provided clear comparisons against distillation-based methods (DHO), showing that co-training handles biased priors better.

### Outstanding Concerns (Reasons for Rejection):
*   Despite the rebuttal, the core criticism remains: the paper is essentially a "targeted recipe." It assembles well-known techniques (Co-training, Mixup, Entropy-weighting, and PEFT) to achieve a performance boost. Reviewer rE4S (R4) explicitly labeled the contribution as "severely overclaimed," arguing that the work merely shifts dependency from labels to high-quality CLIP priors.
*   While the authors defend CaPT as a "general framework," it lacks a truly novel mechanism that would change how the community approaches SSL beyond this specific combination of tools.
*   The approach requires maintaining and training two distinct models, which introduces a complexity burden that may not be justified by the architectural innovation provided.

**Reviewer Scores:**

Reviewer **u1wY (R1)** Moved from a strong reject (2) to a tentative pass after being satisfied with the theoretical clarification and EuroSAT results.

**Uzf4 (R2)** Likely remains at a 4. While the authors addressed the domain-dependence, the concern regarding "careful combination of known pieces" was not structurally resolved.

---

### Decision · Program_Chairs · 2026-01-26

Reject